# Lentiviral vector induces high-quality memory T cells via dendritic cells transduction

Min Wen Ku [1,2,3,4], Pierre Authié[1], Fabien Nevo[1], Philippe Souque [2], Maryline Bourgine [1,2], Marta Romano [5], Pierre Charneau[1,2,6 ✉] & Laleh Majlessi [1,6 ✉]

We report a lentiviral vector harboring the human β2-microglobulin promoter, with predominant expression in immune cells and minimal proximal enhancers to improve vector safety. This lentiviral vector efficiently transduces major dendritic cell subsets in vivo. With a mycobacterial immunogen, we observed distinct functional signatures and memory phenotype in lentiviral vector- or Adenovirus type 5 (Ad5)-immunized mice, despite comparable antigen-specific CD8[+] T cell magnitudes. Compared to Ad5, lentiviral vector immunization resulted in higher multifunctional and IL-2-producing CD8[+] T cells. Furthermore, lentiviral vector immunization primed CD8[+] T cells towards central memory phenotype, while Ad5 immunization favored effector memory phenotype. Studies using HIV antigens in outbred rats demonstrated additional clear-cut evidence for an immunogenic advantage of lentiviral vector over Ad5. Additionally, lentiviral vector provided enhance therapeutic anti-tumor protection than Ad5. In conclusion, coupling lentiviral vector with β2-microglobulin promoter represents a promising approach to produce long-lasting, high-quality cellular immunity for vaccinal purposes.

[1] Laboratoire Commun Pasteur-TheraVectys, Institut Pasteur, Paris, France. [2] Unité de Virologie Moléculaire et Vaccinologie, Institut Pasteur, Paris, France. [3] Université Paris Diderot, Sorbonne Paris Cité, Paris, France. [4] Ecole Doctorale Frontières du Vivant (FdV), Paris, France. [5] Unit In Vivo Models, Sciensano, Brussels, Belgium. [6] These authors jointly supervised this work: Pierre Charneau, Laleh Majlessi. ✉email: pierre.charneau@pasteur.fr; laleh.majlessi@pasteur.fr

A large amount of preclinical and clinical evidence has demonstrated the essential role of T cells in mediating protection against cancers and pathogens[1–4]. Thus, the development of T-cell vaccines has become a major axis of research. So far, T-cell vaccines can induce substantial T-cell frequencies but still fail to achieve long-term efficacy against many chronic diseases[5]. Immunization strategies aiming at T-cell induction face two remarkable challenges: (i) identification of protective T-cell targets and (ii) maintenance of T-cell functionality and protective capacity in a durable manner. To date, immunization with viral vectors remains the most efficient strategy to induce CD8+ T-cell responses, having the edge over other molecular (e.g., DNA, RNA, and peptide) or cellular (e.g., dendritic cell-based, attenuated microorganism) vaccines[6]. Compared to other vaccine strategies, viral vectors efficiently deliver the transgene intracellularly, resulting in a sustained antigen expression. This favors the effective presentation through the major histocompatibility complex-I presentation machinery, initiating CD8+ T-cell responses. Furthermore, viral vector immunization induces local innate immune responses like a natural viral infection, promoting adaptive immune responses, thereby avoiding the need for additional adjuvant.

Multiple viral vectors, including poxviruses, alphaviruses, and adenoviruses-derived vectors, have been developed primarily for vaccination purposes[7–9]. Poxviruses were the first engineered antigen-encoding viral vectors to be tested in clinical trials. These include canarypox virus ALVAC, modified vaccinia virus Ankara (MVA), and New York attenuated vaccinia virus (NYVAC)[9]. However, the usage of poxvirus-derived vectors was impeded by (i) high prevalence of anti-vector immunity in the community[10] and (ii) weak capacity to induce memory T-cell responses[6]. Alphavirus-based vectors are self-limiting, as they provide a high transgene expression level at the cost of detrimental cell toxicity that interferes with antigen expression[11]. The widely adopted adenoviral vectors provide a more potent T-cell immunity compared to poxvirus-derived vectors[6], giving them an advantage over the latter for clinical applications. However, similar to poxvirus-derived vectors, adenoviral vectors also suffer a dampened effectiveness in T-cell immunity due to pre-existing vector-specific immunity in the population[12]. Even though pre-existing immunity against adenoviral vectors can be circumvented by employing serotype variants, the weak immunogenicity induced by these variants prevents them from being the top pick for vaccine vectors[13,14].

Lentiviral vectors emerged as a powerful vaccine and gene therapy platform[15–20], exhibiting several advantages over other viral vectors. Lentiviral vectors are majorly pseudo-typed with heterologous vesicular stomatitis virus envelop glycoprotein (VSV-G), to which human populations have negligible exposure. Therefore, lentiviral vectors do not suffer a decreased immunogenicity from the pre-existing vector-specific immunity. Lentiviral vectors are also capable of transducing dividing and non-dividing cells, notably dendritic cells[21,22], the most potent antigen-presenting cells (APCs) with a unique ability to activate naïve T cells. The ability of lentiviral vectors to transduce APCs ensures prolonged endogenous antigen presentation due to the expression of antigen-encoding genes in the APCs throughout their lifespan. Considering the above-mentioned traits of lentiviral vectors, they offer an attractive option for further vaccine development suited for clinical applications. However, lentiviral-induced T-cell responses lack comprehensive documentation and systematic comparison with T-cell responses induced by other clinically adopted viral vectors.

In the present study, with the perspective of attaining better safety and optimal antigen-expression in relevant immune cells, we first developed a lentiviral vector harboring the human β2-microglobulin (β2m) promoter. We showed that intramuscular (i.m.) immunization with lentiviral vector employing the β2m promoter led to highly efficient in vivo dendritic cell transduction and in vivo transgene expression that lasted for at least 7 days post-immunization (dpi). We then investigated the magnitude, phenotype, functions, and memory features of antigen-specific CD8+ T cells, induced by lentiviral vector immunization in mouse and rat models, in a head-to-head comparison with Ad5. We notably showed that, at similar frequencies of antigen-specific CD8+ T cells, lentiviral vectors triggered a higher proportion of polyfunctional CD8+ T-cell effectors and induced a larger population of CD8+ T cells with central memory phenotype compared to Ad5. Further extension of our comparative immunogenicity study to the preclinical rat model demonstrated a large immunogenic advantage of lentiviral vector over Ad5. In an immune-therapeutic setting, immunization with lentiviral vector harboring β2m promoter provided enhanced protection and survival in tumor-bearing mice compared to Ad5 immunization. Together with the advantage of quasi-absence of pre-existing anti-vector immunity against lentiviral vectors, these insights highlight the strengths of lentiviral vectors in a vaccination setting to achieve robust and durable CD8+ T-cell immunity. These results show that this lentiviral vector system has a potential critical avenue for the development of prophylactic and therapeutic vaccines against a broad range of infectious diseases or malignancies.

## Results

**An engineered lentiviral vaccinal vector to derive transgene expression under the regulation of human β2-microglobulin (β2m) promoter.** Cytomegalovirus (CMV) promoter is commonly used as a viral internal promoter, owing to its high basal activity in a wide range of cell types. However, the presence of enhancer elements in CMV promoters could potentially cause hazardous insertional mutagenesis[23]. Towards better safety and immune response efficiency of lentiviral vector, we investigated the human β2m promoter, which comprises minimal proximal APC-specific enhancer elements and is widely active in immune cells, notably dendritic cells. This promoter consists of highly conserved cis-regulatory elements, designated as interferon-stimulated response elements (ISREs) and SXY modules (Fig. 1a). The ISREs are the binding sites for the interferon regulatory factor family, while the SXY modules are cooperatively bound by a multiprotein complex. Together, these mediators drive the transactivation of the promoter[24]. The mediators of these regulatory elements are tightly coordinated by immune effectors such as cytokines, assuring a high transcriptional activity of this promoter, mainly in immune cells[25]. To assess the efficacy of β2m promoter in dendritic cells, we transduced murine Bone marrow-derived dendritic cells with an integrative lentiviral vector coding for the reporter green fluorescent protein (GFP) at a multiplicity of infection (MOI) of 10, under the regulation of CMV (LV-CMV-GFP) or β2m (LV-β2m-GFP) promoter. In parallel, we also transduced the human embryonic kidney (HEK) 293T cells, a cell line that is efficiently transduced by a lentiviral vector as a positive control. Flow cytometric analysis at day 3 post-transduction of dendritic cells showed that LV-β2m-GFP transduction led to similar percentages of GFP+ cells and mean fluorescence intensities (MFI) when compared to LV-CMV-GFP (Fig. 1b). In HEK 293T cells, the CMV promoter clearly showed better efficiency than the β2m promoter (Fig. 1b). These results demonstrated that the lentiviral vector harboring the β2m promoter—with minimal APC-specific enhancer sequences—leads to high activity in dendritic cells, equivalent to the powerful CMV promoter which contains hazardous enhancer elements.

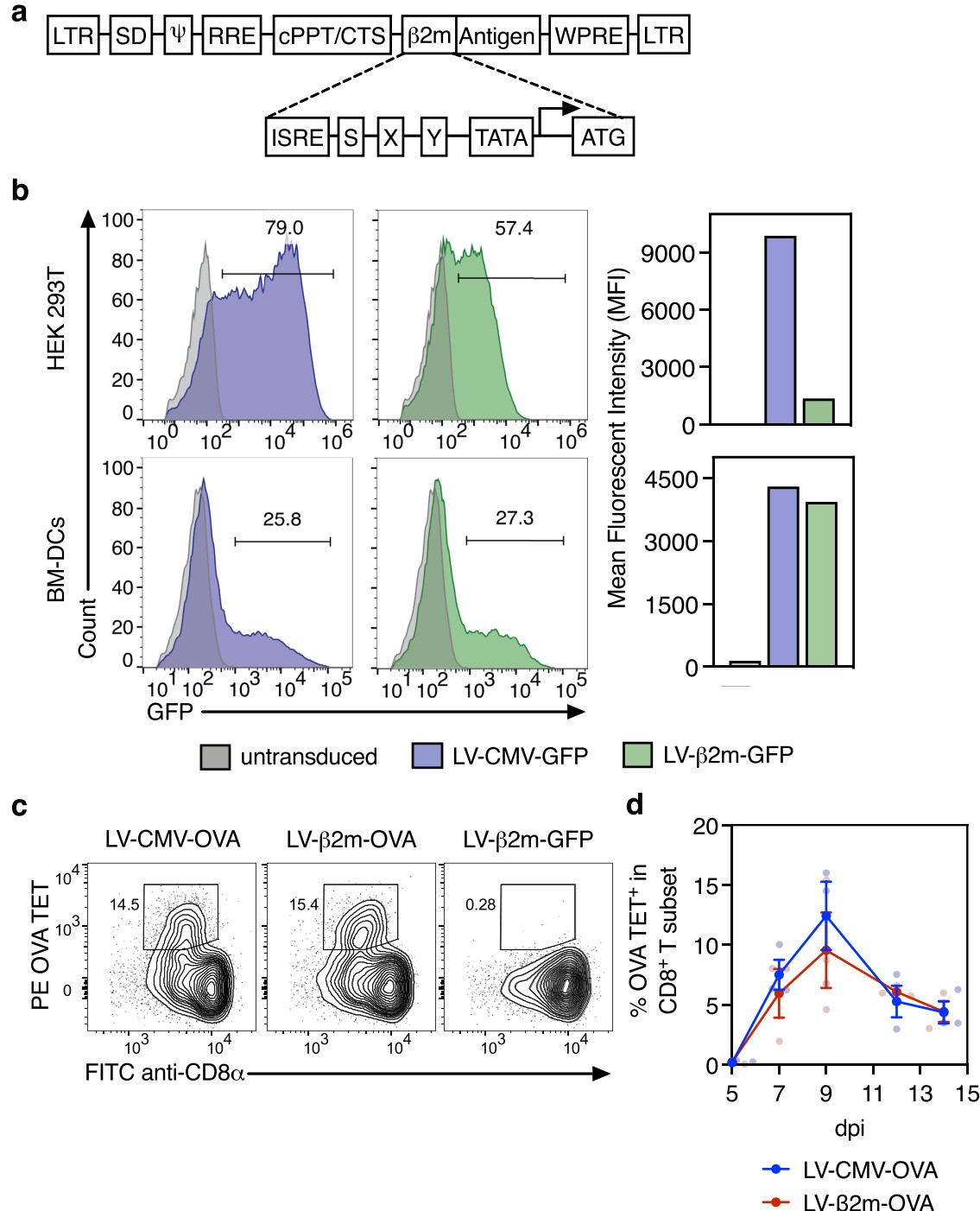

**Fig. 1 Comparison of CMV and β2m promoter in terms of cell transduction efficiency and in vivo CD8+ T-cell induction. a** Schematic representation of the segment of the transfer plasmid, containing β2m internal promoter, used in the lentiviral vector. LTR long terminal repeats; SD, Shine–Dalgarno sequence; RRE, rev response element; cPPT, central polypurine tract; CTS, central termination sequence; ISRE, interferon-stimulated response element; WPRE, woodchuck hepatitis virus posttranscriptional regulatory element. **b** HEK 293T and bone marrow-derived dendritic cells from C57BL/6 mice were transduced with LV-CMV-GFP (blue) or LV-β2m-GFP (green) at MOI 10 for 3 days. Percentage of GFP+ cells (left) and mean fluorescent intensity (right) of GFP+ cells were compared by flow cytometry. **c** C57BL/6 mice ($n = 3$/group) were immunized i.m. with $5 \times 10^7$ TU of LV-CMV or LV-β2m encoding for OVA:242–353 immunodominant, H-2Kb-restricted epitope. Representative cytometric plots for OVA tetramer CD8+ T cells at 9 dpi, with LV-β2m-GFP-immunized mice used as negative controls. **d** The frequencies of OVA-specific CD8+ T cells in the blood determined by cytometry at 5, 7, 9, 12, and 14 dpi, expressed as mean ± SEM. Smaller dots represent biological replicates.

In addition, using the surrogate truncated ovalbumin (OVA:232-343) xenoantigen, we observed that i.m. immunization of C57BL/6 mice ($n = 3$/group) with $5 \times 10^7$ TU (Transduction Unit) of LV-CMV-OVA or LV-β2m-OVA led to the comparable magnitude and kinetics of antigen-specific CD8+ T-cell responses, as monitored using PE-(Kb-SIINFEKL)$_4$ tetramer (OVA TET) in the peripheral blood lymphocytes (PBL) (Fig. 1c, d). We also observed a comparable magnitude of antigen-specific CD8+ T-cell responses in PBL of mice immunized with LV-CMV-OVA or LV-β2m-OVA via the intravenous (i.v.) route (Supplementary Fig. 1a, b). However,

these responses were generally of lower magnitude when compared to the responses obtained after immunization by the i.m. route (Fig. 1c, d, Supplementary Fig. 1a, b). Therefore, the i.m. route of immunization was chosen for further studies.

**Biodistribution of transgene expression after immunization with lentiviral vector harboring CMV or β2m promoter.** To further compare the impact of CMV or β2m promoters on the biodistribution, kinetics, and strength of transgene expression after lentiviral vector immunization, we performed bioluminescence imaging in C57BL/6 mice ($n = 3$/experimental group and $n = 2$/control group) at 3, 5, 7, 9, 12, and 14 days post-immunization (dpi) by i.m. administration of $5 \times 10^7$ TU of the integrative lentiviral vector carrying β2m-nano-Luciferase (nLuc) or LV-CMV-nLuc. The LV-β2m-GFP was used in parallel as a negative control. Immunization with LV-β2m-nLuc or LV-CMV-nLuc led to similar localization of the bioluminescence signal, mainly concentrated at the immunization site at 5 dpi (Fig. 2a). Both LV-β2m-nLuc or LV-CMV-nLuc immunization also led to a relatively short burst of transgene expression, detectable until 7 dpi (Fig. 2b). The strong bioluminescence signal after immunization was confirmed to originate majorly from the muscle (Fig. 2c).

To better characterize the biodistribution of LV-β2m, we conducted a more detailed bioluminescence analysis at earlier time points of 3, 4, 5, 6, 7 and 12 dpi in C57BL/6 mice ($n = 5$/group) injected i.m. with $5 \times 10^7$ TU of LV-β2m-nLuc (Fig. 2d). A strong nLuc signal was consistently detected at the site of the injection from 3 to 5 dpi, with a downward trend commencing at 6 dpi, and regression to the background level at 7 dpi (Fig. 2e). To further understand the transgene persistence in vivo, we applied quantitative real-time PCR (qPCR) to spleen, pLNs, and muscle of lentiviral vector-immunized mice. We detected traces of nLuc transgene in pLNs and muscles until 30 dpi, but not in the spleen. At 90 dpi, nLuc transgene was no longer detectable in these organs, despite the integrative nature of the lentiviral vector used (Supplementary Fig. 2). Therefore, after lentiviral vector immunization, a rapid burst of antigen expression occurs until 6 dpi, after which the transgene persistence until 30 dpi may still allow minute expression of antigen. Altogether, the expression of the antigen encoded by lentiviral vector is no longer detectable from 7 dpi, as determined by the very sensitive imaging technology using nLuc. Subsequently, there is complete clearance of the lentiviral vector from 30 dpi, as determined by qPCR.

**Dendritic cells of all major subsets transduced with lentiviral vector are able to activate CD8+ T cells in vivo.** Since we demonstrated that the transgene expression was mainly localized at the site of i.m. injection, we then focused on the T-cell activating properties of APCs, notably the dendritic cell subsets in the pLNs of lentiviral vector-immunized mice. To identify the immune cell subsets transduced in vivo by lentiviral vector harboring the β2m promoter, C57BL/6 mice ($n = 6$) were immunized i.m. with $5 \times 10^7$ TU of LV-β2m-GFP or LV-β2m-OVA as a negative control, and pLNs were collected at 5 dpi. Immune cell subsets were defined as follows: T cells (CD3+ B220−), B cells (CD19+ B220+), macrophages (CD11b+ F4/80+ Mφs), conventional dendritic cells (CD11c+ CD8− or CD11c+ CD8+), or plasmacytoid dendritic cells (CD11c+ B220+) (Fig. 3a). Various proportions of all these subsets were transduced by the lentiviral vector, with higher percentages of GFP+ cells, i.e., 2–5%, detected in macrophages, conventional dendritic cells and plasmacytoid dendritic cells, versus only 0.47% and 0.76% detected in T and B cell subsets, respectively (Fig. 3b). Such ability of lentiviral vector to transduce a large range of immune cell types was expected

because of their pseudo-typing with VSV-G envelop glycoprotein, which can interact with a broad spectrum of cell types through low-density lipid receptor (LDLR)[26]. The lower infectibility of quiescent B and T cells by the lentiviral vector may result from their weak LDLR expression[27].

We then determined which lentiviral vector-transduced cell subsets were capable of CD8+ T-cell activation. To do so, we immunized i.m. C57BL/6 mice ($n = 3$) with $5 \times 10^7$ TU of LV-β2m-OVA, or LV-β2m-GFP as a negative control, then collected the pLNs at 5 dpi. The pLN cells were first magnetically enriched for CD11c+ population and then electronically sorted into CD11c+ CD8− conventional dendritic cells, CD11c+ CD8+ conventional dendritic cells and CD11c+ B220+ plasmacytoid dendritic cells. The CD11c− cell fraction was further sorted into T cells, B cells, and CD11b+ F4/80+ Mφs (Fig. 3c). Each sorted cell fraction was then co-cultured with CFSE-labeled OVA-specific OT1 TCR transgenic CD8+ T cells, at a ratio of 1 sorted cell to 10 OT1 cells. After 2 days of co-culture, the CFSE dilution in the dividing OT1 cells was measured using flow cytometry, as an indicator of T-cell activation and proliferation. We observed that only CD11c+ CD8− conventional dendritic cells, CD11c+ CD8+ conventional dendritic cells, and plasmacytoid dendritic cells—but not Mφs, B or T cells—induced OT1-cell proliferation (Fig. 3d). Thus, upon i.m. immunization with a lentiviral vector harboring the β2m promoter all the major dendritic cell subsets acquire the capacity to present lentiviral vector-derived antigens and to activate antigen-specific CD8+ T cells.

**Advantage of lentiviral vector over Ad5 at inducing long-lasting polyfunctional CD8+ T cells.** Having selected the β2m promoter for vaccinal lentiviral vector and characterized the in vivo cellular features of T-cell activation after immunization, we then compared the T-cell immunity induced by lentiviral vector to that of the gold standard adenoviral vector serotype 5 (Ad5). As a model antigen, we chose EsxH (TB10.4), an immunogen of vaccinal interest derived from the intracellular pathogen *Mycobacterium tuberculosis*[28,29]. Due to the strong tropism of Ad5 for epithelial cells but not immune cells[30], we used the conventional CMV promoter, rather than β2m promoter in this vector. We also determined the dose of $1 \times 10^7$ IGU (infectious genome unit) of Ad5-CMV as the lowest quantity yielding CD8+ T-cell frequencies comparable to those generated by $5 \times 10^7$ TU of LV-β2m at 28 dpi, using the OVA model antigen (Supplementary Fig. 3a, b). Therefore, we immunized i.m C57BL/6 mice with $1 \times 10^7$ IGU of Ad5-CMV-EsxH or $5 \times 10^7$ TU of LV-β2m-EsxH per mouse, and compared the frequencies of CD8+ T cells specific to the immunodominant EsxH:3-11 epitope in the PBL, from 5 to 90 dpi by using PE-(Kb-IMYNYPAM)4 tetramer (PE EsxH TET) (Fig. 4a). We also followed CD8+ T-cell responses in the spleen at 28 and 90 dpi to assess the early and late memory responses, respectively (Fig. 4b). In mice immunized with either vector, we measured antigen-specific CD8+ T-cell responses of equal magnitude, at any time point, in PBL (Fig. 4a and Supplementary Fig. 4) or in the spleen (Fig. 4b and Supplementary Fig. 4).

We also extensively analyzed the CD8+ T-splenocyte functional responses by intracellular cytokine staining (ICS) at 28 and 90 dpi after ex vivo stimulation with EsxH:3-11 synthetic peptide and determined the frequencies of single, double or triple IFNγ/TNFα/IL-2 positive CD8+ T cells together with the CD107α degranulation marker (Supplementary Fig. 5a). When examining the CD107α+ IFNγ+ CD8+ T cells, Ad5-CMV-EsxH showed a higher percentage of CD107α+ IFNγ+ at 28 dpi than LV-β2m-EsxH, although no statistical significance could be found. We then measured a regression of CD107α+ IFNγ+ CD8+ T cells to

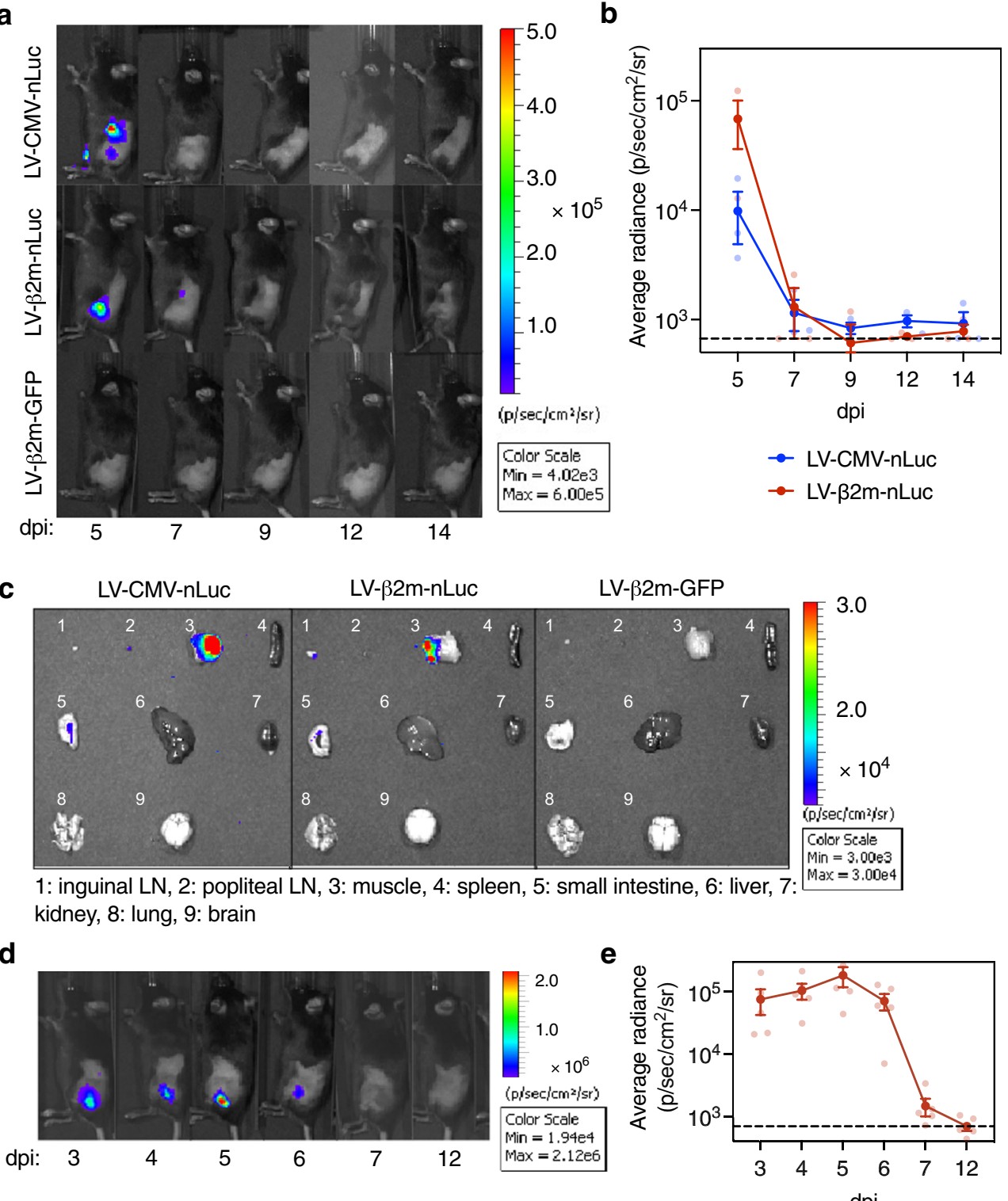

**Fig. 2 Expression kinetics and biodistribution of antigen in mice immunized with LV-CMV-nLuc or LV-β2m-nLuc. a** Lateral view of C57BL/6 mice ($n =$ 3/experimental group and $n =$ 2/control group), immunized i.m. with $5 \times 10^7$ TU of LV-CMV-nLuc, LV-β2m-nLuc, or LV-GFP as a negative control. Furimazine substrate (4 mg/kg) was injected i.p. right before imaging at 5, 7, 9, 12, and 14 dpi. Bioluminescence images indicate the location of nLuc. **b** Mean bioluminescence signal in C57BL/6 mice immunized i.m. with $5 \times 10^7$ TU of LV-CMV or LV-β2m encoding nLuc. **c** Representative bioluminescence images of tissues and organs of C57BL/6 mice ($n =$ 3/experimental group and $n =$ 2/control group) at 5 dpi with $5 \times 10^7$ TU of LV-CMV-nLuc, LV-β2m-nLuc, or LV-GFP as a negative control. LN lymph node. **d** Lateral view of C57BL/6 mice ($n =$ 5) immunized i.m. with $5 \times 10^7$ TU of LV-β2m-nLuc and was imaged at 3, 4, 5, 6, 7, and 12 dpi. **e** The mean bioluminescence signals of mice immunized with LV-β2m-nLuc. For (**b**) and (**e**), the dotted line represents the background level determined from mice immunized with LV-GFP and had received furimazine substrates upon imaging. Each point represents the mean bioluminescence signal (p/s/cm²/sr) per group, expressed as mean ± SEM. Smaller dots represent biological replicates.

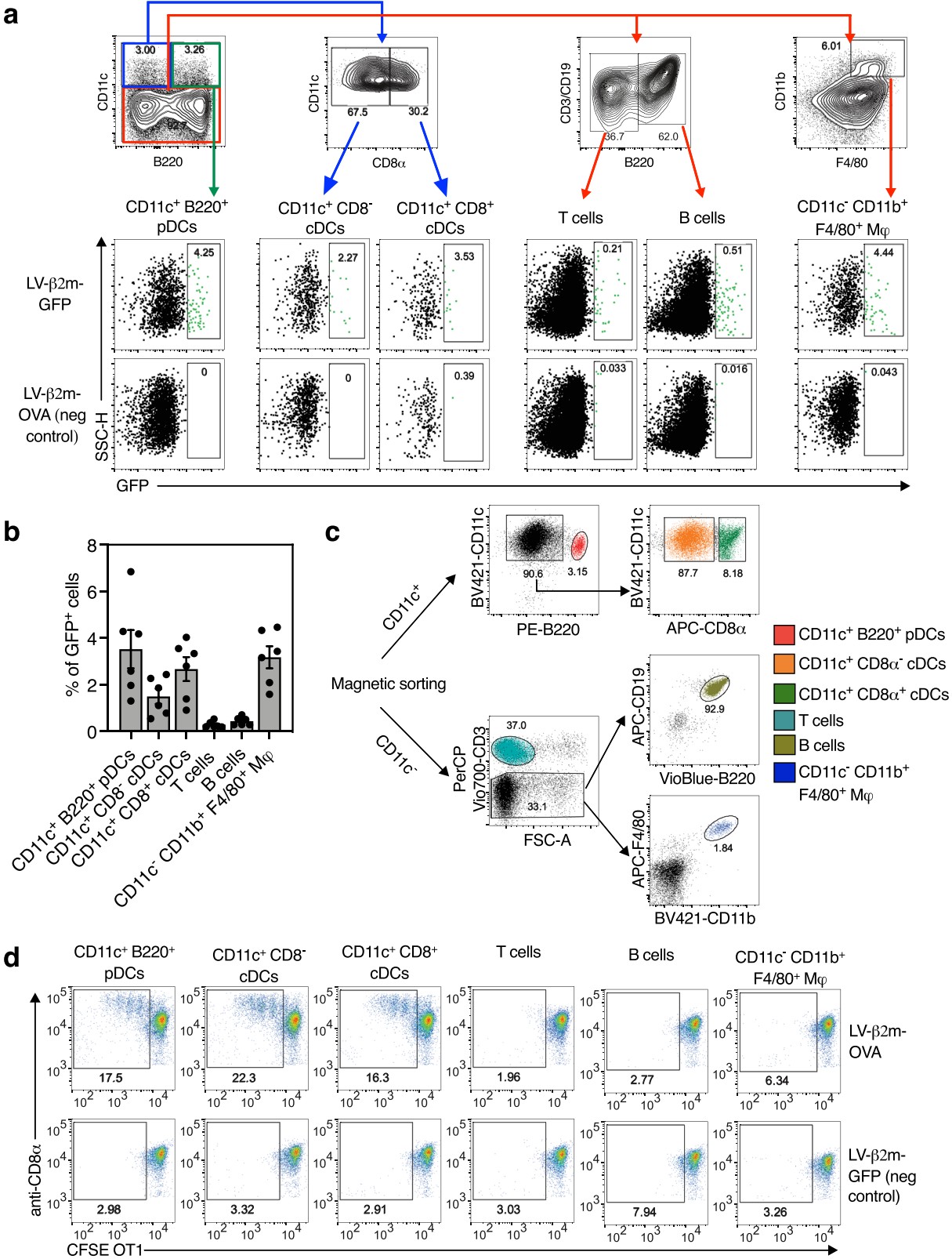

similar levels in both lentiviral vector or Ad5 at 90 dpi (Supplementary Fig. 5b). The distribution of the various functional subsets was globally similar in mice immunized with Ad5-CMV-EsxH or LV-β2m-EsxH, both at early and late time points, with a predominant presence of IFNγ$^+$ TNFα$^-$ IL-2$^-$, IFNγ$^+$ TNFα$^+$ IL-2$^-$ or IFNγ$^+$ TNFα$^+$ IL-2$^+$ cells. However, LV-β2m-EsxH induced a significantly higher percentage of triple positive CD8$^+$ T cells than Ad5-CMV-EsxH, at 28 and 90 dpi (Fig. 5c and Supplementary Fig. 5c). Altogether, these results show that compared to Ad5, the lentiviral vector triggers CD8$^+$ T-cell responses with higher degrees of polyfunctionality, both at the early and late time points post-immunization.

**Fig. 3 Capacity of all major APCs subsets transduced in vivo by lentiviral vector harboring β2m promoter to activate CD8$^+$ T cells. a** C57BL/6 mice ($n = 6$/group) were immunized i.m with $5 \times 10^7$ TU of LV-β2m-GFP or LV-β2m-OVA. Shown are representative cytometric plots for pLN immune cells from an LV-β2m-GFP or an LV-β2m-OVA (negative control) immunized mouse, as analyzed at 5 dpi to detect GFP$^+$ immune cells transduced in vivo by LV harboring β2m promoter. **b** Percentage of GFP$^+$ cells in the respective immune cell subsets expressed as mean ± SEM. Dots represent biological replicates. **c** Sorting strategy for purification of immune cell populations from popliteal lymph nodes of C57BL/6 mice ($n = 3$/group) immunized i.m with $5 \times 10^7$ TU of LV-β2m-GFP or LV-β2m-OVA. **d** CFSE-labeled OT1 CD8$^+$ T cells were cultured for 2 days with the indicated immune cell subsets sorted from C57BL/6 mice immunized with $5 \times 10^7$ TU of LV-β2m-OVA or LV-β2m-GFP (negative control). The activation and cell division of OT1 CD8$^+$ T cells was indicated by the dilution of covalently bound CFSE divided into the daughter cells, as compared between LV-β2m-OVA and LV-β2m-GFP. For (**a**), (**b**), (**c**), and (**d**), pDCs, plasmacytoid dendritic cells; cDCs, conventional dendritic cells; Mφ, macrophages.

## Earlier establishment of T-cell memory phenotype by lentiviral vector compared to Ad5

To get more insights into the long-term T-cell immunity induced by lentiviral vector and Ad5, we comparatively studied the T-cell effector/memory profiles after immunization with Ad5-CMV-EsxH or LV-β2m-EsxH. We cytometrically distinguished antigen-specific EsxH TET$^+$ CD8$^+$ T cells with CD62L$^+$ CD44$^+$ phenotype as central memory T cells ($T_{CM}$) and those with CD62L$^-$ CD44$^+$ phenotype as effector memory T cells ($T_{EM}$) (Fig. 5a). We also studied the expression of CD127 (IL-7 receptor α-chain) and killer cell lectin-like receptor subfamily G member 1 (KLRG1) among the $T_{EM}$ populations to distinguish the potential long-lived memory from the short-lived effector T-cell populations. The antigen-specific CD8$^+$ CD127$^+$ KLRG1$^-$ $T_{EM}$ were designated as memory precursor effector cells (MPECs), while antigen-specific CD8$^+$ CD127$^-$ KLRG1$^+$ $T_{EM}$ were defined as short-lived effector cells (SLECs)[31]. The CD127$^-$ KLRG1$^-$ subset was defined as early effector cells (EECs) (Fig. 5a).

We monitored these populations at (i) the early memory time point of 28 dpi, which falls within the contraction phase of T-cell response, and (ii) the late memory time point of 90 dpi, when the memory population begins to be stabilized[32]. At 28 dpi, LV-β2m-EsxH immunization primed on average higher percentages of antigen-specific CD8$^+$ $T_{CM}$ splenocytes, while Ad5-CMV-EsxH immunization primed higher percentages of antigen-specific CD8$^+$ $T_{EM}$ (Fig. 5b). At 90 dpi, both LV-β2m-EsxH and Ad5-CMV-EsxH groups experienced a decrease in $T_{EM}$ and an increase in $T_{CM}$ frequencies (Fig. 5b). Further cell subsets analysis showed that the $T_{EM}$ induced by LV-β2m-EsxH were distributed evenly among the SLECs, MPECs, and CD127$^+$ KLRG1$^+$ cells, but $T_{EM}$ induced by Ad5-CMV-EsxH were composed of up to 50% of terminally differentiated SLECs (Fig. 5c). Therefore, at 28 dpi, Ad5-CMV-EsxH induced significantly more SLECs, but significantly fewer MPECs than the lentiviral vector (Fig. 5d). At 90 dpi, the percentages of MPECs in Ad5-immunized mice reached those induced earlier and maintained in lentiviral vector-immunized mice (Fig. 5d). We observed no difference in the percentages of EECs or CD127$^+$ KLRG1$^+$ cells at any time point between the two groups. We further reinforced these conclusions by extending the observation to CD8$^+$ T cells specific to OVA antigen, in mice immunized with Ad5-CMV-OVA or LV-β2m-OVA (Fig. 5e). Based on the higher percentages of $T_{CM}$ and MPECs in lentiviral vector-immunized mice at 28 dpi, antigen-specific CD8$^+$ T cells elicited by lentiviral vector immunization displayed faster memory phenotype acquisition than those elicited by Ad5 immunization.

## Immunogenic advantage of lentiviral vector over Ad5 in the rat model

Considering the variable permissiveness of viral vectors and the distinct repertoire of T-cell epitopes presentable by major histocompatibility complex in various animal species, vaccine vector development requires validation in distinct animal models. In these models, the ability to transduce dendritic cells by viral vectors represents an advantage, because dendritic cells are the most potent APCs to prime CD8$^+$ T-cell responses. Hence, we first evaluated the transduction efficiency of Ad5 or lentiviral vector in both

outbred Sprague-Dawley rat and C57BL/6 mouse bone marrow-derived dendritic cells We showed high dendritic cells transduction efficiency with LV-β2m-GFP at MOI of 100 in both rat dendritic cells (34%) and mouse dendritic cells (44%) (Fig. 6a). However, only weak transduction was observed in both rat and mouse dendritic cells when using Ad5, regardless of the choice of promoter and MOI (Fig. 6b). This observation conforms with the previously described poor dendritic cell transduction by Ad5[33].

We then turned to the Sprague-Dawley outbred rat model to comparatively study the immunogenicity of lentiviral vector and Ad5 encoding an HIV poly-antigenic polypeptide, including the p24, nucleocapsid (NC), Pol and Nef antigens, hereafter referred to as LV-β2m-HIV or Ad5-CMV-HIV. Rats ($n = 8$/group) were immunized with $5 \times 10^7$ TU of LV-β2m-HIV or $1 \times 10^7$ IGU of Ad5-CMV-HIV, and IFNγ ELISpot assay was performed using splenocytes collected at 14 dpi. NC and Pol antigens were not immunogenic in these rats, regardless of the vector used, despite their immunogenicity in mice (Fig. 6c). In contrast, p24 and Nef antigens, which were immunogenic in these rats, gave a clear edge to lentiviral vector over Ad5 at inducing IFNγ-producing T-cell effectors (Fig. 6c). The fact that the LV-β2m-GFP negative control vector did not induce T-cell responses against the HIV proteins (P24, NC, and Pol) clearly indicates that the minute amounts of particles injected for immunization do not induce cross-presentation of these lentiviral vector proteins, which could have interfered with the measurement of T-cell responses against the HIV transgenic antigens.

## Therapeutic LV-β2m vaccination provides enhanced tumor protection and survival

To further elucidate the level of protection conferred by LV-β2m, we employed an EG.7 cancer cell line expressing the OVA antigen as a tumor model. C57BL/6 mice ($n = 6$/experimental group and $n = 5$/control vector group) were challenged subcutaneously with $3 \times 10^6$ cells and tumor growth was monitored hereafter. When the tumor volume averaged ~250 mm$^3$ (large tumor), at 8 days post engraftment, these challenged mice were injected i.m. with a single dose of $5 \times 10^7$ TU of LV-β2m-OVA or $1 \times 10^7$ IGU of Ad5-CMV-OVA. As negative controls, mice implanted with a similar amount of EG.7 cells were injected i.m. with control vectors (LV-β2m-GFP or Ad5-CMV-GFP) or phosphate-buffered saline (PBS). Both mock-immunized mice and unimmunized mice displayed rapid growth of tumor throughout the monitoring period of 38 days post-challenge (Fig. 7a). The most pronounced tumor protection and survival were observed in LV-β2m-OVA-immunized mice, whereby on 9 dpi, the average tumor volume decreased to ~320 mm$^3$ (Fig. 7b). Conversely, in Ad5-CMV-OVA-immunized individuals, the average tumor volume remained at ~585 mm$^3$ at 9 dpi (Fig. 7b). These differences in tumor size were observed despite similar levels of CD8$^+$ T cells specific to the immunodominant OVA:323-339 epitope in the PBL when monitored at 9 dpi (Fig. 7c). The percentage of mice that became tumor-free and remained so throughout the experimentation was 83% (5/6) in LV-β2m-OVA-immunized mice, while in Ad5-CMV-OVA-immunized mice it

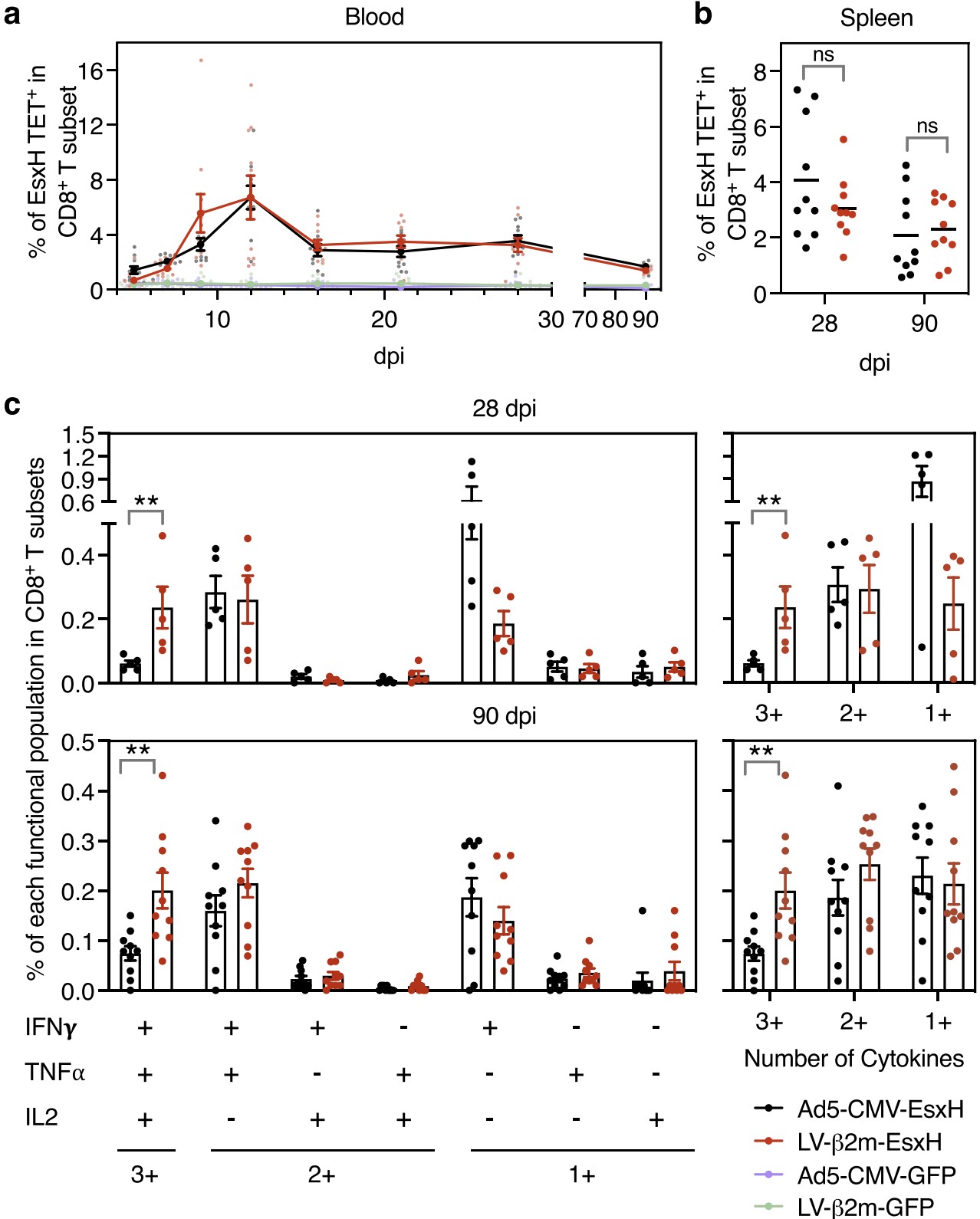

was 50% (3/6) (Fig. 7d). In conclusion, therapeutic immunization with LV-β2m-OVA provides a marked increase in protection and survival in mice compared to Ad5-CMV-OVA.

## Discussion

Vaccination strategies capable of inducing high-quality and sustained T-cell responses are valuable tools in the fight against infectious diseases and cancer[1–4]. Multiple attempts at deploying molecular or cellular vaccine strategies for T-cell induction resulted in modest T-cell responses, insufficient for limiting disease dissemination[5]. The ability of lentiviral vectors to transduce non-dividing cells including dendritic cells is a considerable advantage when it comes to generating excellent antigen-specific immune responses, making it a promising viral vector platform

**Fig. 4 More efficacious induction of polyfunctional T cells by lentiviral vector compared to Ad5 vector.** C57BL/6 mice ($n = 10$/group) were immunized i.m. with $1 \times 10^7$ IGU of Ad5-CMV-EsxH or $5 \times 10^7$ TU of LV-β2m-EsxH, while negative control mice were injected with Ad5-CMV-GFP or LV-β2m-GFP. EsxH:3-11-specific CD8$^+$ T cells were assessed at 28 or 90 dpi by cytometry. **a** Percentages of antigen-specific T cells within the CD3$^+$ CD8$^+$ T subset in the blood, as detected by PE-conjugated (K$^b$-IMYNYPAM)$_4$ tetramer (PE EsxH TET), at the indicated time points post-immunization. Shown are mean ± SEM. Smaller dots represent biological replicates. **b** Percentages of EsxH TET$^+$ within the CD3$^+$ CD8$^+$ T splenocytes of the immunized mice at 28 or 90 dpi. Horizontal lines represent mean. Dots represent biological replicates. **c** Functional profile of EsxH:3-11-specific CD8$^+$ T splenocytes from C57BL/6 mice immunized i.m with Ad5-CMV-EsxH or LV-β2m-EsxH ($n = 5$/group) at 28 dpi (top) or ($n = 10$/group) at 90 dpi (bottom), as investigated by ICS. The quality of response was characterized by the percentages of cells producing every possible combination of the measured cytokines (IFNγ, TNFα, and IL-2). Recapitulative frequencies of single, double, or triple positive CD8$^+$ T cells at 28 or 90 dpi (left). The functional subsets were regrouped based on the number of effector cytokine produced (single (1+), double (2+), or triple (3+) positive cells (right)). Data are presented as mean ± SEM after subtraction of the background observed with the same splenocytes without peptide stimulation. Dots represent biological replicates. Statistical significance was determined via the Mann–Whitney $t$-test (**$p < 0.01$).

for T-cell induction vaccines. However, the currently adopted combination of broad tropism envelope VSV-G and CMV promoter in many lentiviral vectors frequently results in non-targeted transgene expression in all cell types. The pan-cell type enhancer activity of the CMV promoter has shortcomings for translation into clinical usage due to the high risk of insertional mutagenesis[23]. For this reason, we established a targeted transgene expression using the human β2m promoter. The β2m promoter allows for an optimal expression of antigen in restricted subpopulations of APCs with rapid turnover rate and therefore, a short lifespan in the vaccinated host[34], contributing to the safety of the vector.

Besides mouse and rat models, we also demonstrated strong immunity induced by LV-β2m vaccination in macaques. These results, which are now considered for publication, are beyond the scope of the present research but highlight the relevance of this advanced generation of lentiviral vector, at least in non-human primates. More importantly, LV-β2m has demonstrated strong immunogenicity in a recent phase I HIV clinical trial (NCT02054286), establishing its practicality and the absence of adverse effects in humans. Despite our long experience in the field of lentiviral vector-based vaccines in various animal species, we have never observed any safety concern or adverse effect in vaccinated animals, no matter the promoter used, the dose or the route of administration. Such effects have not been reported by any other expert groups either. Therefore, it is difficult to compare the safety of LV-β2m and LV-CMV experimentally in the context of the present work. However, vaccine safety is crucial and the current development of COVID-19 vaccines illustrated further how much any possible adverse effects must be anticipated. Hence, our strategy aiming at limiting the transgene expression to immune cells including APCs by using the β2m promoter instead of the strong ubiquitous CMV enhancer-promoter, with its high probability of insertional mutagenesis[32], can further improve the safety of lentiviral vector.

To examine in vivo the biodistribution of transgene expression after i.m immunization with lentiviral vector harboring the β2m promoter, we used LV-β2 m encoding the nLuc, which has outstanding sensitivity and brightness when compared to the commonly employed Firefly luciferase[35]. We observed that the transgene expression was majorly limited to the site of injection, in this case, the muscle, with detectable bioluminescence signal up to 7 dpi. The duration of transgene expression is dependent on its immunogenicity, such that highly immunogenic antigens could result in shorter-expression due to the elimination of antigen-expressing cells by the immune response. In this case, the minimally immunogenic GFP[36] and luciferase[37] could result in longer transgene expression than the strong *M. tuberculosis* or HIV-derived immunogens. Using a highly sensitive qPCR-based approach, we detected the presence of transgene at least until 30 dpi, but not at 90 dpi. This observation contrasts with previous

reports describing the long persistence of the transgene up to 90; dpi[38,39]. The discrepancy in the duration of the transgene persistence between these previous works and the present study could arise from differences in doses and routes of lentiviral vector immunization.

The CD8$^+$ T-cell responses induced by LV-β2m displayed high degrees of multifunctionality and notably the ability to produce IL-2. In contrast, the Ad5-induced CD8$^+$ T cells were majorly IFNγ$^+$ TNFα$^+$ IL-2$^-$. IL-2 is required for optimal effector cell expansion[40] and is especially important in the settings of diminished CD4$^+$ T cells to maintain CD8$^+$ T-cell proliferative potential via an autocrine loop[41]. In terms of memory maintenance, endogenous IL-2 is critical for CD8$^+$ T-cell memory homeostasis[42], and administration of exogenous IL-2 can emphasize the memory CD8$^+$ T-cell pool[43,44]. The weak ability of Ad5 to induce IL-2-producing CD8$^+$ T cells can also be indicative of partial CD8$^+$ T-cell exhaustion, as described by Wherry et al.[45]. Numerous human or animal studies on HIV/SIV, tuberculosis, and leishmania indicate that multifunctional T cells are the best effectors in controlling pathogen loads[46–49]. In fact, we also demonstrated that better protective efficacy was obtained by LV-β2m compared to Ad5-CMV in a therapeutic immunization with the EG.7 tumor model, which could be also be linked to the high degree of multifunctionality of T cells induced via LV-β2m immunization. The differences in cytokine profiles induced by the lentiviral vector and Ad5 may be explained by the distinct cell types targeted by the two viral vectors, associated with the respective downstream antigen presentation mechanisms and/or inflammatory pathways.

Besides the polyfunctionality of T cells, we characterized the phenotype of CD8$^+$ T cells induced by LV-β2m vaccination as a predictive measure of the quality and magnitude of memory T-cell responses[50]. After vaccination, several functionally distinct subsets of antigen-specific T cells can be defined as MPECs (CD127$^+$ KLRG1$^-$), SLECs (CD127$^-$ KLRG1$^+$) and $T_{CM}$ (CD62L$^+$ CD44$^-$)[31,51]. Using two distinct antigens, we observed an earlier stabilization of $T_{CM}$ populations and a higher amount of MPECs after lentiviral vector immunization compared to Ad5. $T_{CM}$ cells reside in lymphoid tissues and have long-lived characteristics, while MPECs, though adopting effector-like phenotype, confer a high likelihood of contributing to memory populations[52]. Therefore, the higher percentage of $T_{CM}$ and MPECs found after lentiviral vector immunization indicates a better memory response induced by the lentiviral vector. In line with previous studies, our present investigation demonstrated that Ad5 immunization primed a predominately $T_{EM}$ response, and could be attributed to the long-term antigen persistence after Ad5 administration[53,54]. Ad5 immunization provides sustained antigen stimulus resembling persistent infections such as those caused by HIV and CMV. These persisting infections often induce antigen-specific T cells with low CD127 expression and

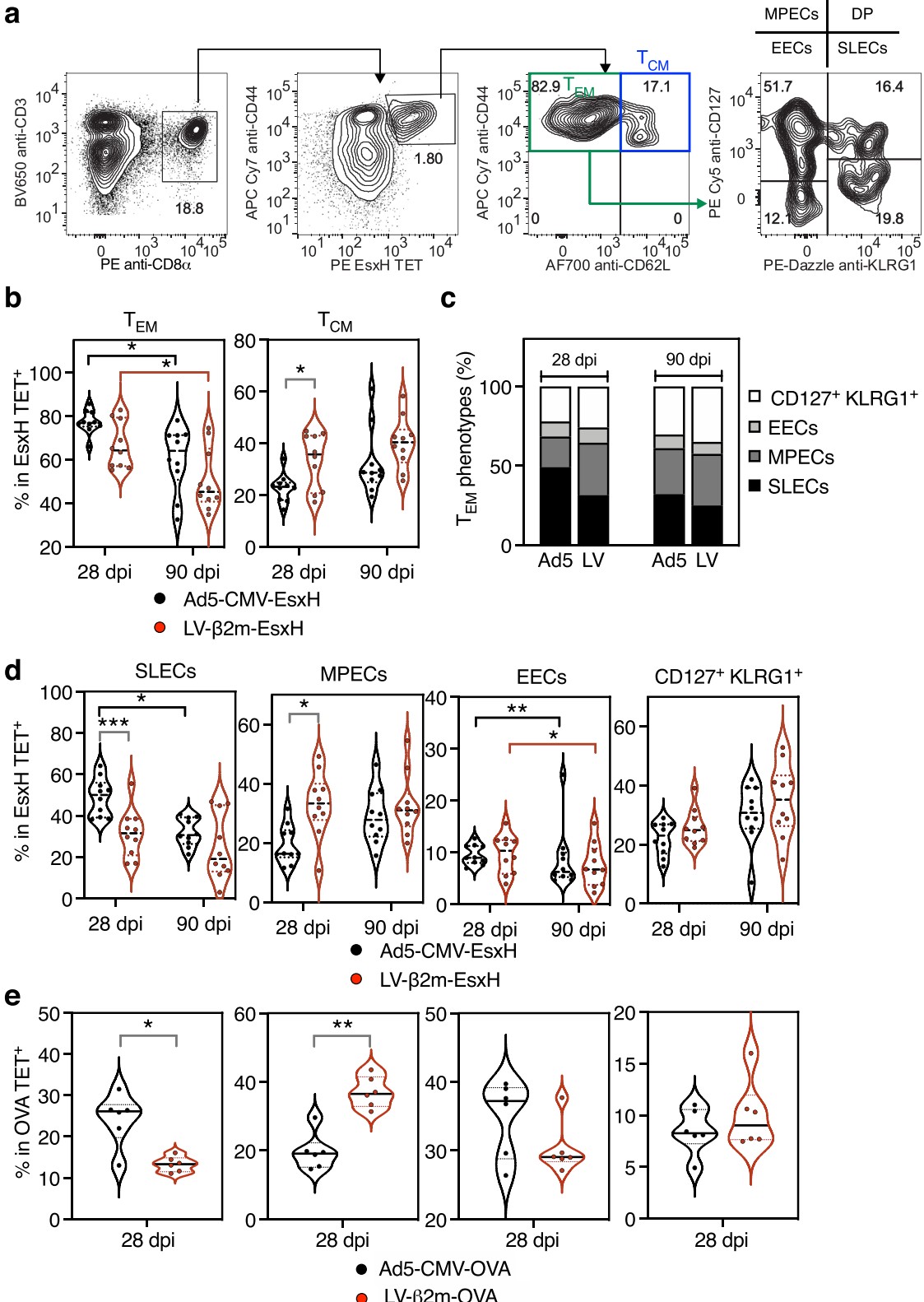

irreversible T-cell dysfunction[55,56]. In contrast, acute lymphocytic choriomeningitis virus (LCMV) and *Listeria monocytogenes* infection have shorter antigen availabilities and induce T cells with high CD127 expression during the effector phase[57,58]. LV-β2m immunization, in this aspect, is rather similar to the latter case where a short burst of antigen expression can give rise to high T-cell memory frequency.

Defining the optimal vaccine dose is a crucial step to reach maximal vaccine efficacy at the requisite safety level. In this study, we selected the lentiviral vector and Ad5 vector doses that gave rise to equivalent amounts of antigen-specific CD8+ T cells. In clinical trials, the Ad5 vector is administered at $5-15 \times 10^{10}$ viral particles per individuals[59]. The dose of Ad5 used in this study, i.e., $1 \times 10^7$ IGU of Ad5/mouse, corresponds to $\approx 10 \times 10^{10}$ viral

**Fig. 5 Earlier commitment to memory T cells after immunization with lentiviral vector compared to Ad5 vector.** C57BL/6 mice ($n = 10$/group) were immunized i.m with $1 \times 10^7$ IGU of Ad5-CMV-EsxH or $5 \times 10^7$ TU of LV-β2m-EsxH. Various profiles of functional antigen-specific CD8$^+$ T subsets were identified and assessed by cytometry at 28 and 90 dpi by use of PE EsxH TET, used together with antibodies specific to CD62L, CD44, CD127, and KLRG1 markers. **a** Gating strategy to distinguish $T_{CM}$ (central memory cells), $T_{EM}$ (effector memory cells), short-lived effector cells (SLECs), memory precursor cells (MPECs), early effector cells (EECs) and CD127$^+$ KLRG1$^+$ double-positive (DP) cells. **b** Violin plots showing the distribution of $T_{EM}$ and $T_{CM}$ at 28 or 90 dpi. **c** Composition of SLECs, MPECs, EECs, and DP within $T_{EM}$ subsets at 28 or 90 dpi. **d** Percentages of SLECs, MPECs, EECs, and DP, as determined at 28 or 90 dpi. For (**b**) and (**d**), statistical significance was determined via Two-way ANOVA with Tukey's multiple comparison test ($^{****}p < 0.0001$, $^{***}p < 0.001$, $^{**}p < 0.01$, $^{*}p < 0.05$). **e** Percentages of SLECs, MPECs, EECs, and DP at 28 or 90 dpi after immunization i.m with $1 \times 10^7$ IGU/mouse of Ad5-CMV-OVA or $5 \times 10^7$ TU/mouse of LV-β2m-OVA. Statistical significance was determined via Mann–Whitney t-test ($^{**}p < 0.01$, $^{*}p < 0.05$). For (**b**), (**d**), and (**e**), the violin plots show the median (black line) and the interquartile range (dashed line). Each dot represents biological replicate. Statistical test value $P < 0.05$ is not indicated on the graphs.

particles per individual, therefore falling within the recommended range. The dose of the lentiviral vector used in this study, i.e., $5 \times 10^7$ TU/mouse, is within the optimal range determined for murine models[20,60]. Unlike Ad5, when translated to non-human primate[19] or human usage (NCT02054286), the optimal lentiviral vector dose does not need to be scaled up based on body weight and has been empirically shown to be within the same range as that used in the murine models. This could suggest that the lentiviral vector immunogenicity is not optimal in preclinical-animal models, generally underestimating the lentiviral vector efficacy.

In addition to the higher quality of T cells generated after lentiviral vector immunization compared to Ad5, one major consideration on viral vectors for clinical usage is the pre-existing vector-specific immunity, which also gives an advantage to the lentiviral vector. The pre-existing vector-specific immunity includes neutralizing antibodies, specific T cells, or type I IFN-activated NK cells, hampering the antigenicity of the vaccine vector by limiting its persistence and/or eliminating the targeted cells before an efficient initiation of specific antigen expression and thus induction of an immune response[61]. To overcome the setbacks of pre-existing neutralizing antibodies, Adenoviral vectors from rare human serotypes or animal-derived serotypes with low seroprevalence across all geographical regions were engineered[62]. Although pre-existing Ad5 neutralizing antibodies do not heavily impede animal-derived serotypes[63,64], utilization of these serotypes should be taken with extra caution due to the uncertain impact of adenoviruses-specific T cells on vector efficacy[61]. Besides, there is also evidence suggesting that these animal-derived Ad serotypes are less immunogenic and protective than human-derived Ad5 serotypes in the absence of pre-existing immunity in animal models[13,14]. Conversely, the VSV-G pseudo-typed lentiviral vector immunization vector presents no risk of specific vector immunity in humans, as VSV is not a natural human pathogen[65]. If repeated administration of lentiviral vector is required in prime-boost vaccination regimens, the impact of anti-envelope immunity induced after prime can be circumvented by the use of heterologous VSV-G protein envelopes for boost, counteracting possible induction of anti-vector immunity[19,66].

Altogether, our results provide a comprehensive understanding of the in vivo biodistribution of the transgene, transferred by immunization with this engineered lentiviral vector harboring the harm-minimized human β2m promoter. We also studied in parallel lentiviral vector and Ad5 to compare the CD8$^+$ T-cell functions and phenotypes, showing that the lentiviral vector is a better inducer of multifunctional CD8$^+$ T cells. In addition, the CD8$^+$ T cells elicited by lentiviral vector immunization displayed an early commitment to memory phenotype, while Ad5 immunization primed predominantly effector phenotype. Our data also provides an updated view of the molecular and cellular characteristics of effector and memory CD8$^+$ T cells induced by LV-β2m vector in mice, with cross-comparison in a more pertinent outbred rat model,

highlighting the compounded advantages of lentiviral vector over Ad5 vector. Lastly, in the context of OVA-expressing EG.7 tumors, we also demonstrated superior protection mediated by therapeutic LV-β2m immunization over Ad5-CMV in mice. The generation of protective and long-lasting T-cell memory is at the heart of numerous vaccination strategies, while safety is another prerequisite of the utmost importance. Without precluding the obligation to perform preclinical safety evaluation for each lentiviral vector-based vaccine, the engineered lentiviral vector platform equipped with the β2m promoter we herein characterized could meet these two major requirements.

## Methods

**Animals.** Female, 5−6-week-old C57BL/6JRj (H-2$^b$) mice were purchased from Janvier Laboratory (Saint-Berthevin, France). The mice were maintained under specific pathogen-free conditions at Institut Pasteur animal facilities and were immunized at 7−10 weeks of age via intramuscular route or intravenous route with $5 \times 10^7$ TU of lentiviral vector or $1 \times 10^7$ IGU of Ad5 in 50 μl of vector diluted in PBS. C57BL/6-Tg(TcraTcrb)1100Mjb/Crl female OT-1 mice were purchased from Charles River (Écully, France). Female CD Sprague-Dawley rats (201–225 g) were purchased from Charles River Laboratories (Saint Germain Nuelles, France). Rats were maintained in the conventional animal facilities of Sciensano and were immunized at 11–14 weeks of age via an intramuscular route with $5 \times 10^7$ TU of lentiviral vector or $1 \times 10^7$ IGU of Ad5 in 50 μl of vector diluted in PBS. All the studies were conducted in accordance with the European and French directives (Directive 86/609/CEE and Decree 87-848 of 19 October 1987), after approval by the Institut Pasteur Safety, Animal Care and Use Committee, under local ethical committee protocol agreement (APAFIS#19863-2019031917064954 and APA-FIS#20289-2019060616332025) or after approval by the Ethical Commission of Sciensano (ethical committee project agreement 20180910-01).

**Lentiviral vectors construction, production, and titration.** The transfer plasmids containing genes coding for EsxH, OVA (242–353), HIV, nLuc, or GFP were constructed by inserting the amplified PCR products into the BamHI and XhoI restriction sites of the pFLAPΔU3-β2m-WPRE or pFLAPΔU3-CMV-WPRE. Each PCR product was inserted between the β2m or CMV promoter and a WPRE (Woodchuck Posttranscriptional Regulatory Element) sequence. The plasmid DNA was verified by restriction digestion and sequencing the region proximal to the transgene insertion sites (Eurofins, Ebersberg, Germany). Plasmids for lentiviral vector production were produced with Macherey-Nagel maxiprep kit (Düren, Germany). Lentiviral particles were produced by transient calcium phosphate co-transfection of HEK-293T cells with: (i) transfer vector pTRIPΔU3β2m plasmid, (ii) expression pHCMV-G plasmid, coding for VSV-G envelope protein of serotype Indiana (IND), and (iii) an encapsidation pNDK plasmid[67]. The crude lentiviral vector stocks were ultracentrifuged, resuspended in sterile PIPES buffer pH 7.5, supplemented with 2.5% glucose, and stocked at −80 °C. The titer of the lentiviral vector was determined via qPCR with HEK-293T cells treated with aphidicolin to block the cell division[18]. Briefly, HEK-293T cells were transduced with lentiviral vector stocks and heat-inactivated vector (30 min at 70 °C) as plasmid contamination control. After 48 h of transduction, genomic DNA was isolated and subjected to qPCR for detection of U5 in the lentiviral vector and CD3 as reference gene. The following primer pairs were used to detect U5: 5′–GCTAGAGATTT TCCACACTGACTAA–3′ and 5′–GGCTAACTAGGGAACCCACTG–3′. The following primer pairs were used to detect CD3: 5′–GGCTATCATTCTTCTTC AAGGTA–3′ and 5′–CCTCTCTTCAGCCATTTAAGTA–3′. The number of lentiviral vector copies per cell was calculated by normalizing the number of U5 copies to the number of CD3 copies, which indicates the number of 293T cells. Lentiviral vector stocks were diluted with PBS before inoculation into mice.

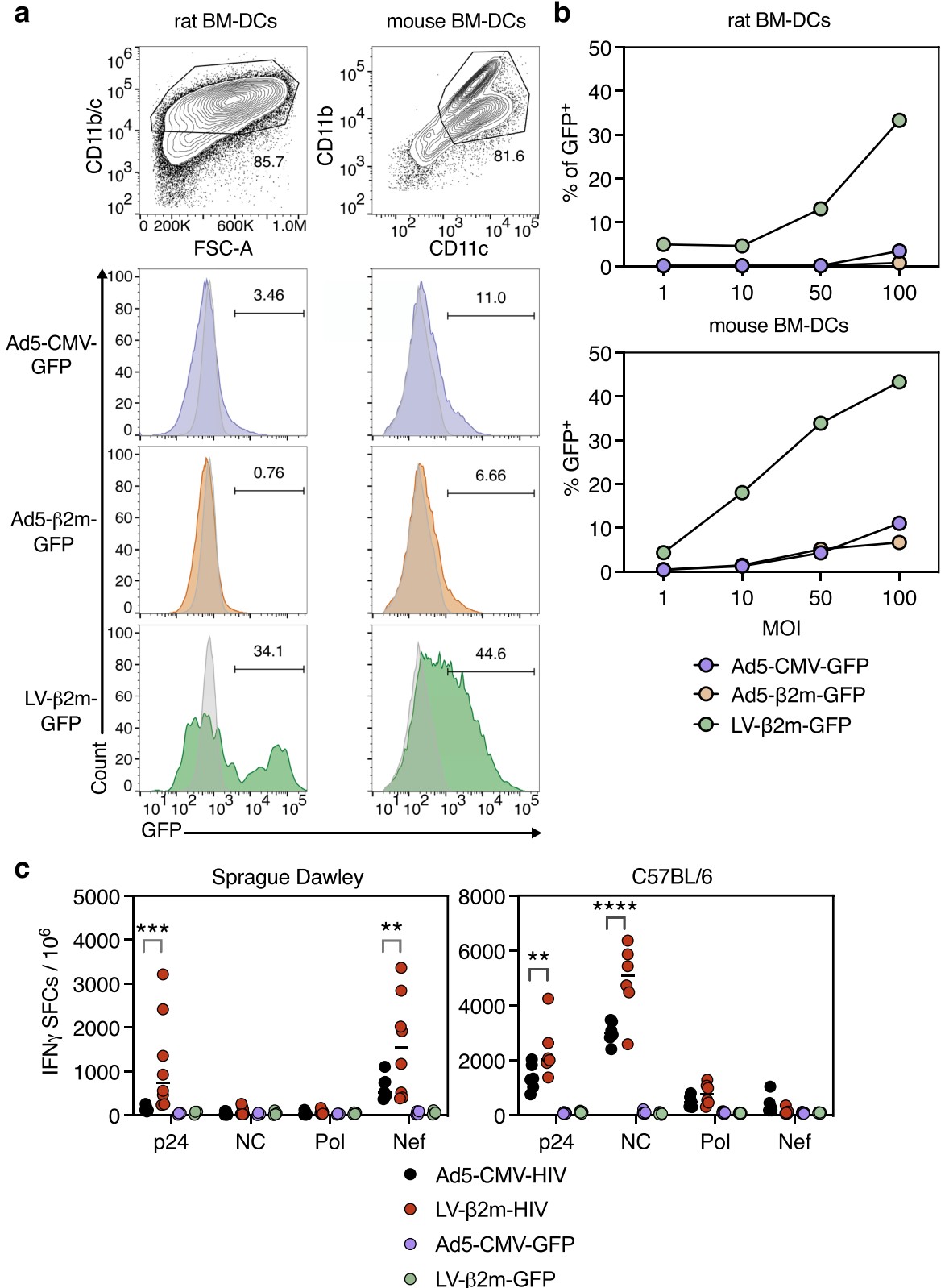

**Adenoviral vector construction, production, and titration**. Ad5 was produced using the ViraPower Adenoviral Promoterless Gateway Expression Kit (Thermo Fisher Scientific, France). The CMV-BamH1-Xho1-WPRE or β2m-BamH1-Xho1-WPRE sequence was amplified from the pFLAPΔU3-β2m-WPRE or pFLAPΔU3-CMV-WPRE by PCR, using the forward primer encoding the attB1 in the 5′ end and reverse primer encoding both the attB2 and SV40 polyA signal sequence in the 5′ end. The attb-PCR product was first cloned into the gateway donor vector pDORN207, via BP Clonase reaction, to form the pDORN207 CMV-BamH1-

Xho1-WPRE-SV40 polyA. The gene of interest was then sub-cloned into the pDORN207 CMV-BamH1-Xho1-WPRE-SV40 polyA plasmid, using the BamH1 and Xho1 restriction sites. To obtain the final Ad5 plasmid, the pDORN207 vector, containing the gene of interest, was further cloned into the pAd/PL-DEST[TM] vector via LR Clonase reaction. The Ad5 viral particles were generated by transfection of the E3-transcomplementing HEK-293A cell line, by the pAd CMV-Ag-WPRE-SV40 polyA plasmid, followed by subsequent vector amplification. Ad5 virions were purified using the Adeno-X rapid Maxi purification kit and

**Fig. 6 Wider immunogenic advantage of lentiviral vector over Ad5 in the rat model. a** Percentage of CD11b/c+ GFP+ bone marrow-derived dendritic cells from Sprague-Dawley rats and CD11c+ CD11b+ GFP+ bone marrow-derived dendritic cells from C57BL/6 mice were analyzed via flow cytometry 3 days after transduction with Ad5-CMV-GFP, Ad5-β2m-GFP, or LV-β2m-GFP at MOI 100. **b** Percentage of CD11b/c+ GFP+ or CD11c+ CD11b+ GFP+ cells from bone marrow-derived dendritic cells of Sprague-Dawley rats and C57BL/6 mice, respectively, after 3 days post-transduction with Ad5-CMV-GFP, Ad5-β2m-GFP or LV-β2m-GFP at MOI 1, 10, 50 and 100. **c** Sprague-Dawley rats ($n = 8$/group) or C57BL/6 ($n = 6$/group) were immunized i.m. with $1 \times 10^7$ IGU of Ad5-CMV-HIV vectors or $5 \times 10^7$ TU of LV-β2m-HIV vectors, and the splenocytes were subjected to IFNγ ELISpot assay using pools of 15-mers, overlapping by 10 a.a., after 14 dpi. Horizontal bars represent means. Each dot represents a biological replicate. Statistical significance was determined via Two-way ANOVA with Tukey's multiple comparison test (***$p < 0.001$, **$p < 0.01$).

concentrated using the Amicon Ultra-4 10k centrifugal filter unit. The final Ad5 virions were resuspended and stocked at $-80\ ^\circ$C in PIPES buffer pH 7.5, supplemented with 2.5% glucose. Ad5 were titrated using qPCR protocol based on Gallaher et al.[68], performed using HEK-293T cells instead of HeLa cells.

**In vivo bioluminescence imaging.** Bioluminescence imaging on live animals was performed using the IVIS Imaging System (IVIS Spectrum, Perkin Elmer) coupled to a charged-couple device camera. Prior to bioluminescence imaging, mice were anesthetized with 2% isoflurane in oxygen and maintained in a control flow of 1.5% isoflurane in oxygen through a nose cone during imaging. The substrate furimazine (Z108) (provided by Dr. Yves Janin, Institut Pasteur) was dissolved at 2 mg/ml in acidic ethanol. Furimazine was further diluted in sterile D-PBS to the desired concentration prior to injection. Mice were shaved at the site of lentiviral vector injection to enhance signals to noise ratio. Furimazine was injected intraperitoneally at 4 mg/kg, then mice were placed in the imaging chamber and imaged 10 min after furimazine administration. Sequential images were captured under the auto-exposure settings. To image individual organs and tissues, mice were dissected 10 min after administration of furimazine at 4 mg/kg and imaged within 3 min after dissection with an exposure time of 60 s. Images from each experimental set were analyzed using Living Image Software (Ver. 2.50.1 Xenogen). Measurements from regions of interest were selected and the luminescence values were evaluated as average radiance (p/s/cm$^2$/sr). The baseline signals were obtained from mice immunized with the control LV-β2m-GFP vector.

**Quantitative PCR (qPCR) for transgene persistence detection.** Total DNA was isolated from 100 mg of spleen, muscles, and pLN using QIAmp DNA Mini Kit and was eluted in 100 μl of elution buffer. The integrated transgene sequence, nLuc, was amplified with the primer pairs 5′–ACCAAATGGGCCAGATCGAAAA–3′ and 5′–CCATAGTGCAGGATCACCTTAAAGT–3′. The control amplifier mouse glyceraldehyde 3-phosphate dehydrogenase (*gapdh*) gene was included for each sample, using the primer pairs 5′–GTTGTCTCCTGCGACTTCA–3′ and 5′–GGTGGTCCAGGGTTTCTTA–3′. Reactions were performed in duplicates on 100 ng of genomic DNA with 5pmol of each primer using the iQ™ SYBR® Green Supermix (Biorad, France) in a total of 20 μl/reaction. The negative control consisted of blank reagents and water. For the positive control, genomic DNA extracted from samples after 5 dpi was used. The qPCR cycle was performed on Eppendorf RealPlex2 qPCR Real-Time PCR ThermoCycler. The following thermal profile was used: a single cycle of polymerase activation for 3 min at 95 °C, followed by 40 amplification cycles of 15 s at 95 °C and 30 s at 60 °C (annealing-extension step). The data from each sample were normalized to values obtained for *gapdh*.

**Lymph node preparation and cell surface staining.** Aseptically removed pLNs were first digested for 30 min in RPMI-1640 media containing 0.2 mg/ml Collagenase IV (Worthington) and 0.1 mg/ml DNase I (BioRad), at 37 °C under constant agitation. The treated pLNs were then gently homogenized using a 1-ml pipette to disrupt the capsule and passage through a 70-μm nylon mesh, centrifuged, and counted using a hemocytometer (Kova International, CA, USA). For cell surface staining, $5 \times 10^6$ cells were resuspended in FACS buffer (PBS containing 1% BSA) and treated with 2.4G2 (BD Biosciences) to block Fc receptor at 4 °C for 20 min. Cells were subsequently stained with the following surface markers at 4 °C for 30 min: VioBlue anti-CD11c (REA754, Miltenyi), PerCP Vio700 anti-CD3 (REA606, Miltenyi), PE-anti-B220 (REA755, Miltenyi), PE-Vio615 anti-F4/80 (REA126, Miltenyi), PE Vio770 anti-CD4 (REA604, Miltenyi), APC anti-CD8 (REA601, Miltenyi) and APC Vio770 anti-CD11b (M1/70.15.11.5, Miltenyi). Cells were washed with FACS buffer before acquisition on Attune NxT flow cytometer (ThermoFisher Scientific, USA).

**Isolation of immune cell populations by fluorescence-activated cell sorting and ex vivo cell proliferation assay.** Naïve CD8+ T cells from OT-1 mice spleen were enriched using the CD8+ T cells isolating kit according to the manufacturer protocol (Miltenyi Biotech, Cambridge, MA). The purified OT-1 CD8+ T cells were stained in 1 μM of cell trace CFSE for 10 min at 37 °C, 5% CO$_2$. MACS-based CD11c enrichment was used to separate the CD11c+ and CD11c− populations from pLNs of lentiviral vector-immunized mice (Miltenyi Biotech, Cambridge, MA) before further sorting into distinct immune cell subsets. CD11c+ fraction was stained with PerCP Vio700-anti-CD3 (REA606, Miltenyi), BV421 anti-CD11c

(clone N418, BD Biosciences) and PE anti-B220 (REA755, Miltenyi) to sort for CD3− CD11c+ CD8− conventional dendritic cells, CD3− CD11c+ CD8+ conventional dendritic cells and CD3− CD11c+ B220+ plasmacytoid dendritic cells. CD11c− fraction was stained with PerCP Vio700-anti-CD3 (REA606, Miltenyi), APC anti-F4/80 (REA126, Miltenyi) and BV421 anti-CD11b (M1/70.15.11.5, BD Biosciences) to sort for CD3+ B220− (T cells) and CD11b+ F4/80+ (macrophages). CD11c− fraction was also stained with PerCP Vio700-anti-CD3 (REA606, Miltenyi), APC anti-CD19 (6D5, Miltenyi) and VioBlue anti-B220 (REA755, Miltenyi) to sort for CD3− CD19+ B220+ (B cells). This staining step was performed for 30 min at 4 °C in PBS containing 0.5% BSA and 2 mM EDTA. Cell sorting was done using the BD FACSAria sorter and keeping the cells at 4 °C throughout the entire process. Immediately after cell collection, $1 \times 10^4$ sorted cells were co-cultured with $1 \times 10^5$ CFSE-stained OT-1 cells in RPMI media supplemented with 100 μg/ml of streptomycin, 100 IU/ml of penicillin, and 10% of heat-inactivated Fetal Bovine Serum (FBS, Gibco) in a 96 flat-bottom well plate. The proliferation of OT-1 cells was analyzed with a cytometer after two days of co-culture.

**Intracellular cytokine staining.** To detect cytokine responses, $1 \times 10^6$ cells splenocytes were cultured in 96-well plates with 10 μg/ml of a synthetic peptide of interest, appropriate dilution of PE-CD107a mAb, 1 μg/ml of anti-CD28 and anti-CD49d (BD Biosciences) mAbs as a moderate polyclonal stimulation, all in complete RPMI 1640-Glutamax (Invitrogen, Cergy Pontoise France). RPMI was supplemented with 100 μg/ml of streptomycin, 100 IU/ml of penicillin, 10% of heat-inactivated FBS, 50 μM β-mercaptoethanol, 1 mM sodium pyruvate, 1X non-essential amino acid, and 20 mM HEPES. After 3 h of incubation, 1 μg/ml of Brefeldin A (Biolegend, France) and 1 μg/ml of monensin (Biolegend, France) were added and the incubation was prolonged for 3 h. For surface staining, the following panel of mAbs was used: PE-anti-CD8 (KT15, Santa Cruz) and Near IR-live dead (Thermofisher). Cells were harvested and stained with the above-mentioned antibodies for 30 min at 4 °C and subsequently washed twice with PBS 1% BSA. For ICS, the following panel of mAbs was used: APC-anti-IFNγ (XMG1.2, BD Biosciences), BV421-anti-IL-2 (JES6-5H4, BD Biosciences), FITC-anti-TNFα (MP6-Xt22, Biolegend), and PerCP Vio700-anti-CD3 (REA606, Miltenyi). After the surface staining, the intracellular cytokine staining was performed using BD cytofix/cytoperm kit (BD Biosciences, France). Briefly, cells were treated with a permeabilizing solution for 20 min at 4 °C and washed twice by PermWash buffer prior to 30 min incubation at 4 °C with appropriate dilutions of anti-cytokine mAbs prepared in the PermWash buffer. Cells were washed twice in this buffer before cell acquisition on an Attune cytometer (Life Technology). Collected data were analyzed with FlowJo Software Version 10.5.3 (Three Star Inc.).

**Antigen-specific T cells staining.** For antigen-specific T cells staining, PE-conjugated (K$^b$-IMYNYPAM)$_4$ tetramer (PE EsxH TET), or (H-2K$^b$)$_4$-SIINFEKL (PE OVA TET) (MBL International, MA, USA) tetramers were used. Tetramer staining of whole blood was performed with TET at room temperature for 10 min before adding the other surface markers: PerCP-Vio700-anti-CD3 (REA606, Miltenyi) and FITC-anti-CD8α (KT15, Santa Cruz) mAbs at 4 °C for an additional 20 min. Tetramer staining of the T splenocytes was performed with TET at room temperature for 10 min before further staining with BV650-anti-CD3 (145-2C11, BD Biosciences), FITC-anti-CD8a (KT15, Santa Cruz), PE-Cy5-anti-CD127 (A7R34, BD Biosciences), PE-Dazzle-anti-KLRG1 (2F1, Biolegend), AF700-anti-CD62L (MEL-14, BD Biosciences), and APC-Cy7-anti-CD44 (IM7, BD Biosciences) mAbs at 4 °C for an additional 20 min incubation. Prior to surface staining, cells were treated with 2.4G2 to block Fc receptors. Cells were washed twice before acquisition.

**IFNγ ELISpot.** Mouse and rat ELISpot kits were purchased from Mabtech AB (Nacka Strand, Sweden). Pre-coated PVDF plates were used, according to the manufacturer's instructions. Splenocytes from immunized C57BL/6 mice were added into the well triplicates at $1 \times 10^5$ cells/well, stimulated with 1 or 2 μg/ml of relevant or negative control peptide. Concanavalin A was used at 1 μg/ml as T-cell activation control. Unstimulated cells were also included. After 24 h, spots were revealed according to the manufacturers' instructions and counted using an AID ELISpot Reader System ELR04 (Autoimmune Diagnostika GmbH, Strassberg, Germany). Splenocytes from immunized Sprague-Dawley rats were added to the well in quadruplicates at 0.5 or $1 \times 10^5$ cells/well, stimulated with 1 μg/ml of

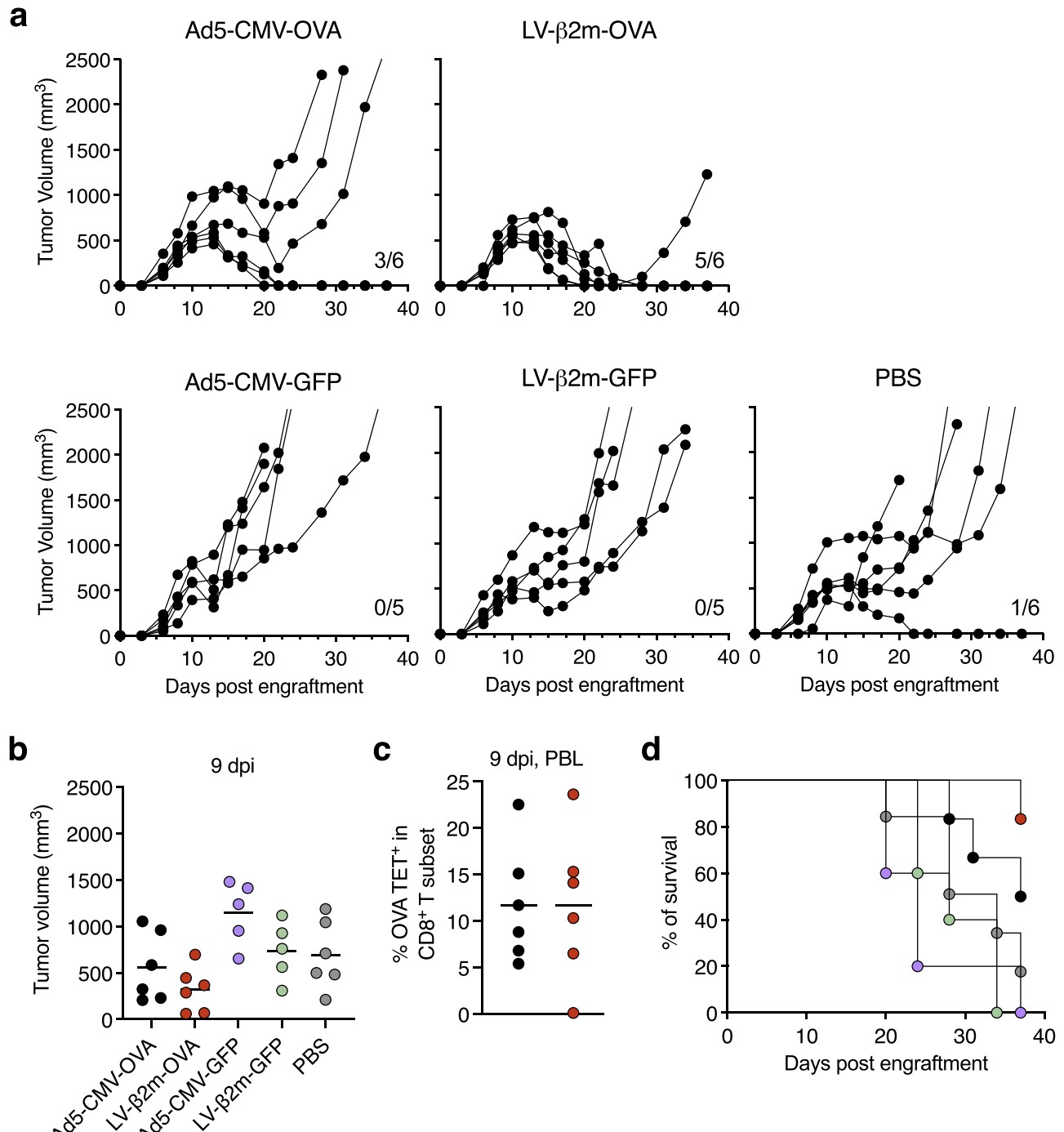

**Fig. 7 Superior protection and survival provided by LV-β2m-OVA therapeutic immunization.** C57BL/6 mice ($n = 6$ for Ad5-CMV-OVA, LV-β2m-OVA, and PBS, $n = 5$ for Ad5-CMV-GFP and LV-β2m-GFP) were challenged subcutaneously with $3 \times 10^6$ E.G7 tumor cell line and were immunized with $5 \times 10^7$ TU of LV-β2m-OVA or $1 \times 10^7$ IGU Ad5-CMV-OVA at day 8 post engraftment when tumor volume averaged ~250 mm$^3$. **a** The tumor volume for individual mouse overtime for the indicated immunization. **b** The tumor volume at 9 dpi (or 17 days post engraftment). **c** Percentages of antigen-specific T cells within the CD3$^+$ CD8$^+$ T subset in the blood at 9 dpi. For (**b**) and (**c**), black lines represent mean, dots represent biological replicates. **d** The percentage of survival at the indicated time points is shown.

relevant peptide mixes. Concanavalin A was used at 2 μg/ml as T-cell activation control. Unstimulated cells were also included. After 24 h, spots were revealed according to the manufacturers' instructions and counted using an ImmunoSpot S6 Ultra-V Analyzer (CTL Europe GmbH, Bonn, Germany). Results were expressed as spot-forming cells per million of splenocytes and the IFNγ secretion background in mock-stimulated wells (or negative control peptide) were subtracted.

**Evaluation of transduction rate in HEK 293 T cells, mouse bone marrow-derived dendritic cells, and rat bone marrow-derived dendritic cells.** HEK 293

T cells were cultured in DMEM media supplemented with 100 μg/ml of streptomycin, 100 IU/ml of penicillin, and 10% heat-inactivated FBS one day before transduction. Cells were transduced with MOI 10 of lentiviral vector and subjected to flow cytometry analysis 3 days after transduction. Mouse bone marrow-derived dendritic cells were prepared from the hematopoietic precursors recovered from femurs of C57BL/6 mice in complete RPMI containing 2% v/v of supernatant of J774 cells stably transfected with murine GM-CSF[69]. Rat bone marrow-derived dendritic cells were prepared from femurs of Sprague Dawley rats and cultured in GM-CSF (30 ng/ml, Peprotech) and IL-4 (5 ng/ml, R&D system). After 7 days of culture, 2 × 10$^6$ of a mouse or rat bone marrow-derived dendritic cells were plated per well in 12-

well plates and transduced by lentiviral vector or Ad5 for 3 days at MOI of 100. Cells were harvested and subjected to flow cytometry analysis to detect GFP$^+$ cells.

**Tumor challenge**. Mice were anesthetized with 2% isoflurane and maintained in a control flow of 1.5% isoflurane in oxygen through a nose cone during injection of EG.7 cells (ATCC® CRL-2113™). Cells were subcutaneously implanted on the right flank in 100 μl of PBS at the indicated dose. When tumor volumes averaged ~250 mm$^3$ after the tumor implantation, mice were randomized into groups of 6 and were immunized with lentiviral vector or Ad5 at the indicated dose. Tumor size was monitored every other day using a digital caliper and tumor volume was calculated using the following formula: $V = (L \times W \times W)/2$, where $V$ is tumor volume, $W$ is tumor width, $L$ is tumor length. Mice were euthanized when the tumor volume reached 1500 mm$^3$, defined as a humane endpoint, approved by APAFIS #20981-2019060616411273.

**Statistics and reproducibility**. Results were expressed as mean and error bars represent SEM. Mann–Whitney tests were used to compare differences when two independent groups were compared. When comparing more than two independent groups, Kruskal–Wallis tests followed by Tukey's multiple comparisons were used. All statistical analysis was performed using Prism v8.01 (GraphPad Software, Inc.). The sample size for each experimental group is detailed in the corresponding figure legend.

**Reporting summary**. Further information on research design is available in the Nature Research Reporting Summary linked to this article.

## Data availability
The authors declare that all data supporting the findings are available within the article and the supplementary figures. The source data for main and supplementary figures are provided as Supplementary Data. All other data are available upon reasonable request from the corresponding author.

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

## Acknowledgements

The authors would like to give additional thanks to Aude Ytebrouck et Fabienne Jurion at Sciensano for their assistance on the Sprague-Dawley rats-related experiments. The authors would also like to thank Yves Janin for providing the furimazine substrate for in vivo bioluminescence imaging. The in vivo bioluminescence imaging was performed at the UtechS Photonic BioImaging (PBI) platform, a member of the France Life Imaging network (grant ANR-11-INBS-0006). M.W.K. was part of the Pasteur—Paris University (PPU) International Ph.D. Program. This project has received funding from the Institut Carnot Pasteur Microbes & Santé, and the European Union's Horizon 2020 research and innovation program under the Marie Sklodowska-Curie grant agreement No 665807. This work is also supported by the Programs Transversaux de Recherche (PTR) # 52-17 from Institut Pasteur.

## Author contributions

Study concept and design: M.W.K., L.M., P.C., acquisition of data: M.W.K., P.A., F.N., M.B., M.R., analysis and interpretation of data: M.W.K., P.A., F.N., L.M., statistical analysis: M.W.K., technical or material support: P.A., P.S., F.N., drafting of the manuscript: M.W.K., L.M.

## Competing interests

P.C. is the founder and CSO of TheraVectys. M.W.K., P.A., and F.N. are employees of TheraVectys. Other authors declare no competing interests.
