## [Peer Review File · Communications Biology]

Reviewers' Comments:

Reviewer #1:

Remarks to the Author:

The manuscript by Ku and coworkers describes the in vivo use of LVs as a vaccine to promote a robust and durable T cell response. They demonstrate that in head-to-head comparisons in mice and rats LV-based vaccine outperforms the 'gold-standard' Ad5 by inducing a polyfunctional and long-lived immune response. The research presented is interesting and the paper is well-written. Please consider following comments and suggestions:

(1) The use/generation of the immune-cell specific B2M promoter is intriguing. In addition to its ability to drive expression in cells (Fig1) it would be interesting to see how it would perform compared to CMV (or other promiscuous promoters) if LVs were administered i.v. Surely, pantropic expression would not be seen, as alluded to by the authors. Current i.m. immunization strategy doesn't highlight/justify, in my opinion, the need to use a cell-specific promoter.

(2) The use of viral vectors, especially in gene therapy, has highlighted the importance of utilising the right model to assess these vectors in to achieve meaningful results pertaining to human physiology (e.g. receptor profiles of target cells in humans vs models of interest). Could the authors comment on how translatable the results would be? Would it be possible to show that the immunization would be as effective through ex vivo experiments/humanized models?

(3) The study investigates the magnitude, phenotype, and function features of CD8+ T cells induced following single dose of LV immunization in great detail. However, there is no evidence shown on how protective this immunization strategy would be against a challenge and if a boost strategy could prime the immune system better.

Minor comments:

1 In the abstract (page 2 line 20) I would be wary about calling the LV system 'novel' as it is a VSV.G pseudotyped HIV-based lentiviral vector commonly used by many researchers. The use of B2M promoter is the novelty presented.

2 Figure 1a, the LV genome schematic is incomplete (i.e lacks 3' LTR)

Reviewer #2:

Remarks to the Author:

In this study, Ku et al report a lentiviral (LV) vector system in which expression of vaccine antigens is driven by the β 2-microglobulin (β 2m) promoter. The authors conduct comparative studies in mice and rats assessing transduction of immune cells and induced T cell responses using the LV- β 2m system and compare these to responses obtained using a CMV promoter based LV vector, and / or an adenoviral vector. This is an interesting study, but the data are comparative rather than demonstrating mechanism. Experiments are generally well performed. There are issues that should be addressed:

A major focus of the paper is the use of an alternative promoter to the commonly used CMV promoter to improve safety of the LV system, thus increasing its appeal as a vaccine vector technology. However, although the risks of insertional mutagenesis using the CMV promoter are briefly discussed, the paper offers no actual evidence that β 2m represents a safer option. This should, at the very least, be discussed in more detail.

The measurement of transgene expression in vivo using luciferase based imaging using the two LV systems has the potential to be informative, however the data shown in Figure 2a and 2b are not convincing. There are only three animals per group, and there appears to be significant variability between individuals in the LV-CMV group. In general, signal seems to be barely above background after the first time point – and the rapid decrease in signal with LV technology seems surprising. It is difficult to determine transduction of specific organs or tissues without harvesting these and imaging separately at individual time points (hence 2b might not be accurate). Alternatively, an

experiment such as the one shown in Supp. Fig 1 could be conducted – this could be performed for the comparison LV-CMV and LV- β 2m also. What is the signal behind the ears of mice transduced with LV-CMV?

The comparison of immunogenicity between LV and Ad5 is complex because the two systems are very different. However, it would be informative to know where the doses selected for each platform sit on a dose response curve, at least in mice. Supp. Fig. 2 suggests that the Ad5 dose (at least with OVA) is at plateau, but what about LV? A discussion of how the two doses selected for the study relate to a typical human dose using these platforms could also be helpful.

In general the claims of this article tend to be overstated. It is debatable that the article demonstrates an 'advantage' of LV over Ad5 in terms of immunogenicity. In rats, a 'clear-cut' advantage of LV over Ad5 is claimed, despite the authors conducting only one experiment, with one antigen. CD8+ T cell frequencies between LV and Ad5 seem comparable in mice using OVA and EsxH as vaccine antigens. The difference in memory phenotype between T cells induced with LV and Ad5 is interesting, but the relevance of this for a vaccine application has not been demonstrated. Challenge studies demonstrating a difference in protective efficacy or differences in the ability of each platform to prime a subsequent boost response (either homologous or heterologous boost) would be informative in this respect. In both mice and rats, LV vectors expressing HIV antigens do seem more immunogenic compared to Ad5 against some epitopes (Figure 6), which is interesting since the LV used in this study appears to be HIV-1 based? Since there is no significant HIV specific response in the LV-GFP control there does not seem to be any significant contribution from delivery of LV virions. Is LV immunogenicity still better in rats if both vectors encode EsxH or is this observation specific to HIV antigens?

The authors readily dismiss other vaccine platforms that are more advanced than LV from a clinical perspective. Some of the comments related to vaccine platforms other than LV are inaccurate. For instance, the authors state that non-human 'animal-derived Ad serotypes are less immunogenic than human-derived Ad5 serotypes' but do not clarify that most of these studies have been performed in mice. It is unclear how non-human adenoviruses would compare to Ad5 in humans without pre-existing immunity to Ad5, but to describe T cell responses induced by these vectors in humans as 'weak immunogenicity' is inaccurate.

Grammar could be improved in places

Point-to-Point Answers

Reviewer #1 lentiviral vector:

The manuscript by Ku and coworkers describes the in vivo use of LVs as a vaccine to promote a robust and durable T cell response. They demonstrate that in head-to-head comparisons in mice and rats LV-based vaccine outperforms the 'gold-standard' Ad5 by inducing a polyfunctional and long-lived immune response. The research presented is interesting and the paper is well-written. Please consider following comments and suggestions:

We thank the reviewer #1 for the constructive criticisms and questions. As detailed below, we performed several experiments to include all requested results. We revised and rephrased the interpretations and conclusions accordingly.

(1) The use/generation of the immune-cell specific B2M promoter is intriguing. In addition to its ability to drive expression in cells (Fig1) it would be interesting to see how it would perform compared to CMV (or other promiscuous promoters) if LVs were administered i.v.

Surely, pantropic expression would not be seen, as alluded to by the authors. Current i.m. immunization strategy doesn't highlight/justify, in my opinion, the need to use a cell-specific promoter.

We performed an additional experiment to address the comparative performance of LV-CMV and LV- β 2m when administered i.v. We immunized i.v. C57BL6JR/j ($n = 5/\text{group}$) with 5×10^7 TU of the respective LV and tracked the antigen-specific CD8⁺ T cells in the blood. We noticed that the magnitude of antigen-specific CD8⁺ T cells in the blood after i.v. administration was lower than that observed after i.m. injection. Subsequent to i.m. injection, we observed 10 – 15 % of antigen-specific CD8⁺ T cells, versus 2 – 4 % after i.v. immunization. However, no statistically significant differences were observed between LV-CMV and LV- β 2m in this framework. The results are now added in Supplementary Fig. 1a-d and commented in Results, Page 5, Lines: 110-115.

It is quite possible that, compared to i.v. immunization, a local administration, such as i.m., s.c. or i.d., more specifically involves local APCs and therefore justifies less the use of a cell/tissue-specific promoter. However, the possibility that LV reaches other cell types in vivo even after local administration cannot be excluded. The CMV promoter is ubiquitous and drives high transgene expression in all cell types. Therefore, its usage, even via local immunization, can be hazardous. Notably, the skeletal muscle is composed of several cell types potentially transducible by LV. Therefore, adopting a cell/tissue-specific promoter ensures more precise and localized antigen production and warrants further safety, even when LV is delivered via i.m. route.

(2) The use of viral vectors, especially in gene therapy, has highlighted the importance of utilizing the right model to assess these vectors in to achieve meaningful results pertaining to human physiology (e.g. receptor profiles of target cells in humans vs models of interest). Could the authors comment on how translatable the results would be? Would it be possible to show that the immunization would be as effective through ex vivo experiments/humanized models?

A now added to Discussion: Page 11; Lines: 272-276: "Besides mouse and rat models, we also demonstrated strong immunity induced by LV- β 2m vaccination in macaques. These results, which are now considered for publication, are beyond the scope of the present research but highlight the relevance of this new generation of LV, at least in non-human primates. More importantly, LV- β 2m has demonstrated strong immunogenicity in a recent phase I HIV clinical trial (NCT02054286), establishing its practicality and the absence of adverse effect in human."

(3) The study investigates the magnitude, phenotype, and function features of CD8⁺ T cells induced following single dose of LV immunization in great detail. However, there is no evidence shown on how protective this immunization strategy would be against a challenge and if a boost strategy could prime the immune system better.

We thank the reviewer for this question. We now performed comparative immuno-onco-therapy by LV- β 2m-OVA and Ad5-CMV-OVA in C57BL/6 mice challenged with OVA-expressing EG.7 cells

which form solid tumors in vivo. Data, included in Figure 7, and detailed/commented in Abstract (Line 28-29), Introduction, Page 4, Lines:77-78, Results, Page 10, Lines: 242-257, Discussion, Page 14, Lines 364-366, Material and Methods, Page 20, Lines: 532-540, Legend to the Figure 7, established that therapeutic immunization with LV- β 2m-OVA provides higher protection and better survival in mice than Ad5-CMV-OVA.

Minor comments:

(1) In the abstract (page 2 line 20) I would be wary about calling the LV system ‘novel’ as it is a VSV.G pseudotyped HIV-based lentiviral vector commonly used by many researchers. The use of B2M promoter is the novelty presented.

Abstract: Page 2, Line 20: We removed “novel” from this sentence.

(2) Figure 1a, the LV genome schematic is incomplete (i.e. lacks 3’ LTR)

Figure 1a: We completed the schematic of LV genome and added LTR at its 3’ end.

Reviewer #2 virologist (Remarks to the Author):

In this study, Ku et al. report a lentiviral (LV) vector system in which expression of vaccine antigens is driven by the β 2-microglobulin (β 2m) promoter. The authors conduct comparative studies in mice and rats assessing transduction of immune cells and induced T cell responses using the LV- β 2m system and compare these to responses obtained using a CMV promoter-based LV vector, and / or an adenoviral vector. This is an interesting study, but the data are comparative rather than demonstrating mechanism. Experiments are generally well performed. There are issues that should be addressed:

We thank the reviewer #2 for constructive questions and advice, which contributed to largely improve the manuscript.

(1) A major focus of the paper is the use of an alternative promoter to the commonly used CMV promoter to improve safety of the LV system, thus increasing its appeal as a vaccine vector technology. However, although the risks of insertional mutagenesis using the CMV promoter are briefly discussed, the paper offers no actual evidence that β 2m represents a safer option. This should, at the very least, be discussed in more detail.

We fully agree with this remark; we added the following paragraph to:

Discussion, Page 11, Lines: 269-274:

“The pan-cell type enhancer activity of CMV promoter has significant shortcomings for translation into clinical usage due to the high probability of insertional mutagenesis occurrence³². For this reason, we established a targeted transgene expression using the human β 2m promoter. The β 2m promoter allows an optimal expression of antigen in restricted subpopulations of APCs that has rapid turnover rate and therefore, a short lifespan in the vaccinated host³³, which can contribute to the safety of the vector.”

And Discussion, Page 11, Lines: 276-284:

“Despite our long experience in the field of LV-based vaccines in various animal species, we have never observed any safety concern or adverse effect in vaccinated animals, no matter the promoter used, the dose or the route of administration. Such effects have not been reported by any other expert groups either. Therefore, it is difficult to compare the safety of LV- β 2m and LV-CMV experimentally in the context of the present work. However, vaccine safety is crucial and the current development of COVID-19 vaccines illustrated further how much any possible adverse effects must be anticipated. Hence, our strategy aiming at limiting the transgene expression to immune cells including APCs by using the β 2m promoter instead of the strong ubiquitous CMV enhancer-promoter, with its high probability of insertional mutagenesis³², can further improve the safety of LV.”

(2) The measurement of transgene expression in vivo using luciferase-based imaging using the two LV systems has the potential to be informative, however the data shown in Figure 2a and 2b are not convincing. There are only three animals per group, and there appears to be significant variability between individuals in the LV-CMV group. In general, signal seems to be barely above background

after the first time point – and the rapid decrease in signal with LV technology seems surprising. It is difficult to determine transduction of specific organs or tissues without harvesting these and imaging separately at individual time points (hence 2b might not be accurate). Alternatively, an experiment such as the one shown in Supp. Fig 1 could be conducted – this could be performed for the comparison of LV-CMV and LV- β 2m also. What is the signal behind the ears of mice transduced with LV-CMV?

We agree with the reviewer that it is difficult to determine the transduction of specific organs by performing only whole-body imaging. Therefore, we performed new imaging experiments, including more animals, also harvesting specific tissues and organs separately to illustrate the signal at these specific localizations at 5dpi. The data are now included in Fig. 2c, and the text is adapted accordingly (Results, Page 6, Lines: 124-125) and Materials and Methods, Page 16, Lines 424-426).

The signal behind the ears of mice transduced with LV-CMV corresponded to background autofluorescence signal. We have now re-analyzed the data and modify the scale of radiance intensity to remove the background signal from both LV-CMV and LV-GFP mice.

(3) The comparison of immunogenicity between LV and Ad5 is complex because the two systems are very different. However, it would be informative to know where the doses selected for each platform sit on a dose response curve, at least in mice. Supp. Fig. 2 suggests that the Ad5 dose (at least with OVA) is at plateau, but what about LV? A discussion of how the two doses selected for the study relate to a typical human dose using these platforms could also be helpful.

The optimal dose of LV in the murine models is $2-5 \times 10^7$ TU/mouse, which has been previously established for various antigens (<https://doi.org/10.1016/j.ymthe.2020.05.016>, <https://pubmed.ncbi.nlm.nih.gov/16308885/>, <https://doi.org/10.1038/sj.mt.6300135> and <https://doi.org/10.1371/journal.pone.0048644>).

To clarify this point, we added the following paragraph to the Discussion, Page 13, Lines: 330-339.

“Defining the optimal vaccine dose is a crucial step to reach maximal vaccine efficacy at the requisite safety level. In this study, we selected the LV and Ad5 vector doses that gave rise to equivalent amounts of antigen-specific CD8⁺ T cells. In clinical trials, the Ad5 vector is administered at $5 - 15 \times 10^{10}$ viral particles (vp) per individuals⁵⁹. The dose of Ad5 used in this study, i.e., 1×10^7 IGU of Ad5/mouse, corresponds to $\approx 10 \times 10^{10}$ vp per individuals, therefore falling within the recommended range. The dose of LV used in this study, i.e., 5×10^7 TU/mouse, is within the optimal range determined for murine models^{20,60}. Unlike Ad5, when translated to non-human primate¹⁹ or human usage (NCT02054286), the optimal LV dose does not need to be scaled up based on body weight and has been empirically shown to be within the same range as that used in the murine models. This could suggest that the LV immunogenicity is not optimal in preclinical-animal models, generally underestimating the LV efficacy.”

(4) In general the claims of this article tend to be overstated. It is debatable that the article demonstrates an ‘advantage’ of LV over Ad5 in terms of immunogenicity. In rats, a ‘clear-cut’ advantage of LV over Ad5 is claimed, despite the authors conducting only one experiment, with one antigen. CD8⁺ T cell frequencies between LV and Ad5 seem comparable in mice using OVA and EsxH as vaccine antigens. The difference in memory phenotype between T cells induced with LV and Ad5 is interesting, but the relevance of this for a vaccine application has not been demonstrated. Challenge studies demonstrating a difference in protective efficacy or differences in the ability of each platform to prime a subsequent boost response (either homologous or heterologous boost) would be informative in this respect.

As also asked by reviewer #1, we now performed comparative immuno-onco-therapy by LV- β 2m-OVA and Ad5-CMV-OVA in C57BL/6 mice against OVA-expressing EG.7 cells which establish solid tumors in vivo. Please see the answer to the question N° 3 of the Reviewer #1.

(5) In both mice and rats, LV vectors expressing HIV antigens do seem more immunogenic compared to Ad5 against some epitopes (Figure 6), which is interesting since the LV used in this study appears to be HIV-1 based? Since there is no significant HIV specific response in the LV-GFP control there does not seem to be any significant contribution from delivery of LV virions. Is LV immunogenicity still better in rats if both vectors encode EsxH or is this observation specific to HIV antigens?

Unfortunately, based on our previous experiments in Sprague Dawley rats, we know that there are no MHC-I-restricted T-cell epitopes on EsxH antigen presentable by these animals (our unpublished observation). To answer to this question, we added to Results, Page 10, Lines: 235-239:

“The fact that the LV-β2m-GFP negative control vector did not induce T-cell responses against the HIV proteins (P24, NC and Pol) clearly indicates that the minute amounts of particles injected for immunization do not induce cross-presentation of these LV proteins, which could have interfered with the measurement of T-cell responses against the HIV transgenic antigens.”

(6) The authors readily dismiss other vaccine platforms that are more advanced than LV from a clinical perspective. Some of the comments related to vaccine platforms other than LV are inaccurate. For instance, the authors state that non-human ‘animal-derived Ad serotypes are less immunogenic than human-derived Ad5 serotypes’ but do not clarify that most of these studies have been performed in mice. It is unclear how non-human adenoviruses would compare to Ad5 in humans without pre-existing immunity to Ad5, but to describe T cell responses induced by these vectors in humans as ‘weak immunogenicity’ is inaccurate.

We re-write and modified the text, Discussion, Page 13-14, Lines: 346-351 to answer to this question: “To overcome the setbacks of pre-existing neutralizing antibodies, Ad vectors from rare human serotypes or animal-derived serotypes with low seroprevalence across all geographical regions were engineered⁶². Although pre-existing Ad5 neutralizing antibodies do not heavily impede animal-derived serotypes^{63,64}, utilization of these serotypes should be taken with extra caution due to the uncertain impact of Ad specific T cells on vector efficacy⁶¹. Besides, there is also evidence suggesting that these animal-derived Ad serotypes are less immunogenic and protective than human-derived Ad5 serotypes in the absence of pre-existing immunity in animal models^{13,14}.”

(7) Grammar could be improved in places

We have corrected the grammatical mistakes, they are indicated in blue in the text.